# Stem cell-derived cranial and spinal motor neurons reveal proteostatic differences between ALS resistant and sensitive motor neurons

Disi An[1†], Ryosuke Fujiki[2,3,4,5†‡], Dylan E Iannitelli[1†], John W Smerdon[6], Shuvadeep Maity[1,7], Matthew F Rose[2,3,5,8,9,10,11], Alon Gelber[2,3,11], Elizabeth K Wanaselja[1], Ilona Yagudayeva[1], Joun Y Lee[2,3§], Christine Vogel[1,7], Hynek Wichterle[6], Elizabeth C Engle[2,3,4,5,11,12,13,14], Esteban Orlando Mazzoni[1,15]*

[1]Department of Biology, New York University, New York, United States; [2]Department of Neurology, Boston Children's Hospital, Boston, United States; [3]FM Kirby Neurobiology Center, Boston Children's Hospital, Boston, United States; [4]Department of Neurology, Harvard Medical School, Boston, United States; [5]Medical Genetics Training Program, Harvard Medical School, Boston, United States; [6]Department of Physiology and Cellular Biophysics, Columbia University Medical Center, New York, United States; [7]Center for Genomics and Systems Biology, New York University, New York, United States; [8]Department of Pathology, Brigham and Women's Hospital, Boston, United States; [9]Department of Pathology, Boston Children's Hospital, Boston, United States; [10]Department of Pathology, Harvard Medical School, Boston, United States; [11]Broad Institute of MIT and Harvard, Cambridge, United States; [12]Howard Hughes Medical Institute, Chevy Chase, United States; [13]Department of Ophthalmology, Boston Children's Hospital, Boston, United States; [14]Department of Ophthalmology, Harvard Medical School, Boston, United States; [15]NYU Neuroscience Institute, NYU Langone Medical Center, New York, United States

*For correspondence:
eom204@nyu.edu

[†]These authors contributed equally to this work

Present address: [‡]Department of Neurology, Kokura Memorial Hospital Kitakyushu, Fukuoka, Japan; [§]Department of Genetics, Albert Einstein College of Medicine, New York, United States

Competing interests: The authors declare that no competing interests exist.

**Abstract** In amyotrophic lateral sclerosis (ALS) spinal motor neurons (SpMN) progressively degenerate while a subset of cranial motor neurons (CrMN) are spared until late stages of the disease. Using a rapid and efficient protocol to differentiate mouse embryonic stem cells (ESC) to SpMNs and CrMNs, we now report that ESC-derived CrMNs accumulate less human (h)SOD1 and insoluble p62 than SpMNs over time. ESC-derived CrMNs have higher proteasome activity to degrade misfolded proteins and are intrinsically more resistant to chemically-induced proteostatic stress than SpMNs. Chemical and genetic activation of the proteasome rescues SpMN sensitivity to proteostatic stress. In agreement, the hSOD1 G93A mouse model reveals that ALS-resistant CrMNs accumulate less insoluble hSOD1 and p62-containing inclusions than SpMNs. Primary-derived ALS-resistant CrMNs are also more resistant than SpMNs to proteostatic stress. Thus, an ESC-based platform has identified a superior capacity to maintain a healthy proteome as a possible mechanism to resist ALS-induced neurodegeneration.

DOI: https://doi.org/10.7554/eLife.44423.001

## Introduction

Neurodegenerative diseases are characterized by the selective death of specific neuronal cell types. Although fundamental to understanding neurodegeneration and designing effective therapies, the molecular nature of differential sensitivity to neurodegeneration remains obscure. Amyotrophic lateral sclerosis (ALS) is a fatal neurodegenerative disease characterized by progressive muscle denervation and loss of motor neurons leading to paralysis (*Brown and Al-Chalabi, 2017*). However, not all motor neurons are affected equally. While most spinal motor neurons (SpMNs) degenerate during ALS progression, a subset of cranial motor neurons (CrMNs: oculomotor, trochlear and abducens motor neurons) are typically spared, allowing patients to retain eye movement until late stages of the disease (*Gizzi et al., 1992*; *Okamoto et al., 1993*; *Lawyer and Netsky, 1953*; *Kanning et al., 2010*). Oculomotor and trochlear motor neurons are also more resistant than SpMNs to neurodegeneration in the ALS mouse model expressing human (h) SOD1 G93A (*Haenggeli and Kato, 2002*). Understanding the intrinsic neuronal mechanisms that contribute to differential sensitivity of motor neurons will deepen our understanding of ALS pathology and ultimately lead down to a path to more effective therapies.

With shared clinical manifestations, ~90% of patients suffer from sporadic ALS while ~10% of ALS cases are familial. More than fifty genes involved in different biological processes have been reported to contribute to the familial ALS cases. (*Rosen, 1993*; *Taylor et al., 2016*). Consequently, many ALS pathogenic mechanisms have been proposed to contribute to motor neuron degeneration (*Taylor et al., 2016*). One common hallmark found in both familial and sporadic ALS is the accumulation of aggregated proteins, which is also shared by many other neurodegenerative diseases (*Yerbury et al., 2016*). ALS mutations in *SOD1*, *TARDBP* (TAR DNA-binding protein 43 kDa), *FUS* (Fused in Sarcoma) and *C9ORF72* generate proteins with a propensity to misfold and aggregate (*Bruijn et al., 1998*; *Neumann et al., 2006*; *Deng et al., 2010*; *Zu et al., 2013*). However, ALS protein inclusions contain more than just ALS-causing proteins. In both ALS patients and mouse models, SpMNs and astrocytes typically contain inclusions positive for ubiquitinated proteins and the ubiquitin-binding protein *SQSTM1* (Sequestosome 1, also known as p62) (*Leigh et al., 1991*; *Watanabe et al., 2001*; *Mizuno et al., 2006*; *Neumann et al., 2006*; *Gal et al., 2007*). Thus, the inability to prevent the accumulation of insoluble protein aggregates could contribute to SpMN sensitivity to ALS.

The ubiquitin proteasome system and autophagy are the two major quality control pathways to maintain proteostasis. Soluble and misfolded proteins are degraded primarily by the ubiquitin proteasome system, while large protein aggregates are recognized and removed by the autophagy lysosome pathway (*Dikic, 2017*). Therefore, ALS-causing misfolding proteins, like mutant SOD1, can be degraded by both the proteasome and autophagy pathways (*Kabuta et al., 2006*; *Castillo et al., 2013*). Moreover, mutations in genes encoding important components of these degradation pathways can cause ALS (*Taylor et al., 2016*; *Ghasemi and Brown, 2018*), including *UBQLN2* (*Deng et al., 2011*), *SQSTM1* (*p62*) (*Fecto et al., 2011*; *Rubino et al., 2012*; *Hirano et al., 2013*), *OPTN* (*Maruyama et al., 2010*), *VCP* (*Johnson et al., 2010*), *VAPB* (*Nishimura et al., 2004*) and *TBK1* (*Freischmidt et al., 2015*). Taken together, this evidence suggests that ALS-sensitive SpMNs are under proteostatic stress during ALS progression (*Atkin et al., 2008*; *Matus et al., 2013*; *Hetz and Mollereau, 2014*).

Previous studies comparing vulnerable and resistant cell types in ALS rodent models have employed laser-capture coupled with RNA level measurements to isolate genes that could contribute to differential motor neuron sensitivity (*Kaplan et al., 2014*; *Allodi et al., 2016*; *Morisaki et al., 2016*). Matrix metallopeptidase 9 (MMP-9) is expressed only in the fast-fatigable α-motor neurons, a selective subtype of SpMNs vulnerable in ALS. Reduction of MMP-9 significantly delayed muscle denervation of fast-fatigable α-motor neurons in the ALS mouse model expressing hSOD1 G93A (*Kaplan et al., 2014*). Conversely, expression of IGF-2 (insulin-like growth factor 2) is upregulated in the resistant oculomotor neurons. Viral delivery of IGF-2 to the muscles of hSOD1 G93A mice extended life-span by 10% (*Allodi et al., 2016*). These studies demonstrate that intrinsic mechanisms influence ALS sensitivity in motor neurons. While manipulating MMP-9 and IGF-2 signaling partially rescues ALS phenotypes, their mechanisms and possible convergent modes of action to resist neurodegeneration are still unknown. Studies attempting to understand the mechanisms of

CrMN resistance to ALS have been mostly limited by difficulties in obtaining large homogenous populations of CrMNs.

Embryonic stem cell (ESC)-based differentiation strategies offer a viable alternative to generate disease-relevant cell types for 'disease in a dish' studies. Traditionally, ESCs can be differentiated into different neuronal subtypes by the activity of signaling molecules. Such is the case with differentiating ESCs exposed to retinoic acid and hedgehog signaling that further differentiate into SpMNs. However, there is no equivalent strategy to derive the developmentally distinct CrMNs at an efficiency conducive for biochemical studies (*Wichterle et al., 2002*). Direct programming of terminal motor neuron fate by transcription factor expression might circumvent this issue. During embryonic differentiation, Isl1 and Lhx3 transcription factors specify SpMN identity (*Thaler et al., 2002*), while Isl1 and Phox2a are essential for CrMN differentiation (*Morin et al., 1997*; *Hirsch et al., 2007*). These different developmental strategies have inspired CrMN and SpMN programming from ESC and human pluripotent stem cells at high efficacy and speed (*Lee et al., 2012*; *Mazzoni et al., 2013*; *De Santis et al., 2018*; *Allodi et al., 2019*).

In this work we employed SpMN and CrMN direct programming strategies to investigate a possible mechanism to explain their differential sensitivity to ALS. We found that resistant CrMN subtypes have fewer insoluble hSOD1 aggregates than SpMNs in an ALS mouse model expressing hSOD1 G93A. By over-expressing hSOD1 mutant proteins, ESC-derived induced CrMNs (iCrMNs) and SpMNs (iSpMNs) recapitulated the differential accumulation of insoluble hSOD1 in vitro and revealed more efficient proteasomal degradation of misfolded proteins in iCrMNs than SpMNs. As a result, iCrMNs also accumulate less insoluble p62 than iSpMNs over time, which is reciprocally validated in the hSOD1 G93A mouse model showing that resistant CrMN subtypes contain fewer p62 positive inclusions than SpMNs. Moreover, iCrMNs were intrinsically more resistant than iSpMNs to proteostatic stress caused by chemicals interfering with endoplasmic reticulum (ER) function. Confirming the predictive value of this ESC-based model, primary CrMNs from the ALS-resistant oculomotor and trochlear nuclei also survived better than primary SpMNs to chemically induced proteostatic stress. Finally, we found that iCrMNs contain more proteasome 20S core subunits and have a higher level of proteasome activity than iSpMNs. Chemical and genetic activation of the proteasome significantly increased iSpMN survival under proteostatic stress. Therefore, our comparative study of ESC-derived iCrMNs and iSpMNs identified that CrMNs take better advantage of the ubiquitin proteasome system to degrade misfolded proteins and maintain proteostasis. This superior proteostatic capacity could be an intrinsic and disease relevant mechanism underlying CrMN resistance to ALS.

## Results

### Reduced hSOD1 accumulation in ALS-resistant motor neurons

To investigate if the ALS-resistant CrMNs maintain a healthier proteome than SpMNs, we compared aggregation of misfolded hSOD1 within these populations in the SOD1 G93A mouse model. To reduce variability from perfusion and staining artifacts, we compared oculomotor/trochlear motor neurons (O/TrMNs) and lumbar SpMNs from segments 4–5 (L4-5) within the same mice. Samples were immunostained with an antibody that specifically recognizes human misfolded SOD1 protein (C4F6) (*Urushitani et al., 2007*; *Bosco et al., 2010*) and an antibody against choline acetyltransferase (ChAT) to identify motor neurons in postnatal day 100 (P100) SOD1 G93A mice.

Interestingly, we found that a third of all SpMNs (34 ± 4.2% [mean ± SEM]; 195 motor neurons from three mice) contain aggregates of misfolded hSOD1 while, by contrast, we were unable to detect any aggregation within O/TrMNs (328 motor neurons from three mice) (*Figure 1A and B*). Additionally, O/TrMNs exhibited strikingly lower overall levels of misfolded hSOD1 compared to SpMNs, necessitating imaging under 5–10 times higher confocal laser intensity. Imaging the SpMNs at the same intensity produced an image with fully saturated motor neurons (*Figure 1—figure supplement 1A*). This result is consistent with the recent observation of little or no aggregation in eye-innervating CrMNs in the G85R SOD1YFP mice (*Thomas et al., 2018*). These data suggest that ALS-resistant CrMNs might be more capable than SpMNs of clearing misfolded hSOD1 and thus mitigating ALS-induced proteostatic stress. To further investigate this possibility, we sought to establish

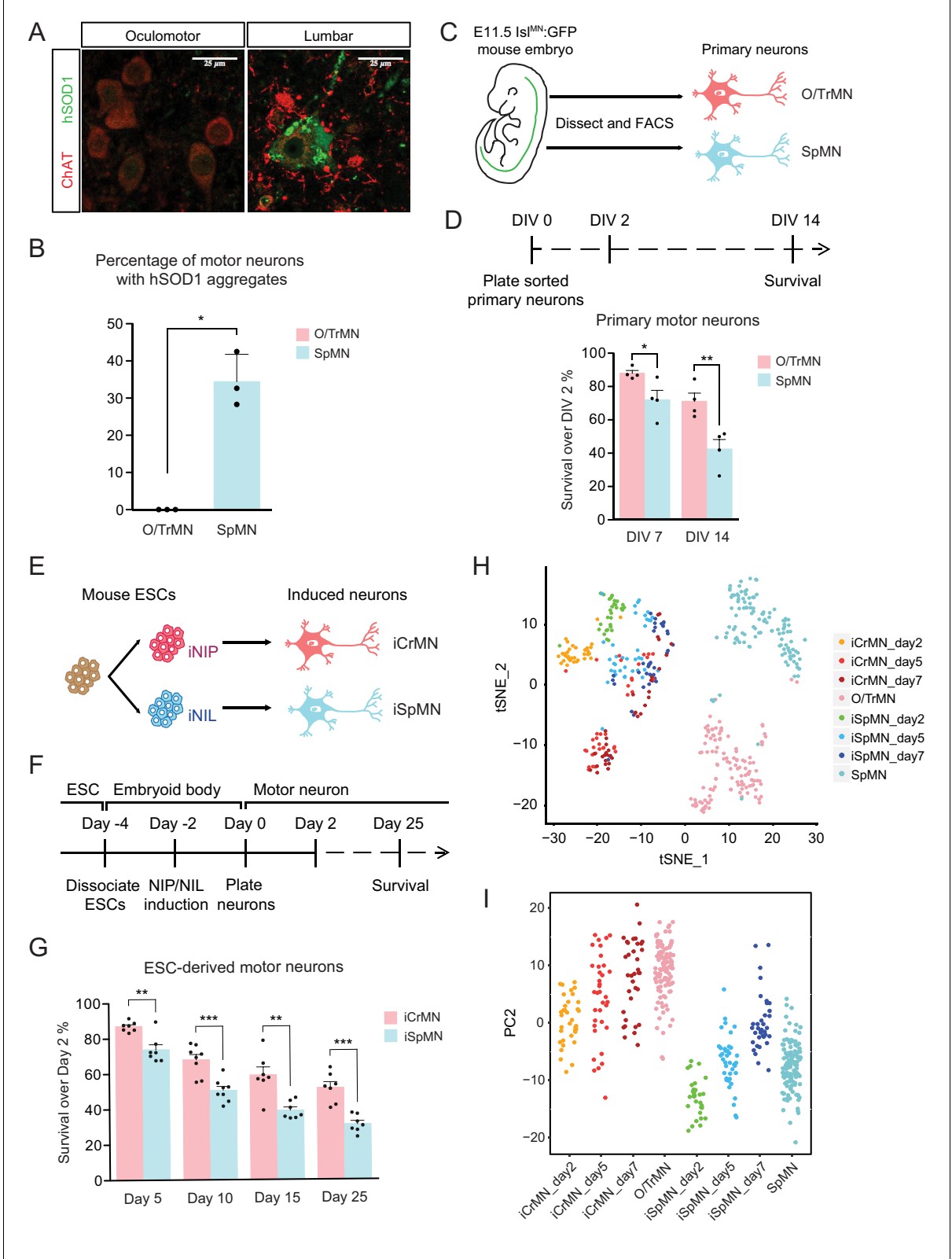

**Figure 1.** ESC-derived iCrMNs and iSpMNs resemble molecular and cellular features of primary O/TrMNs and SpMNs. (**A**) Representative images of oculomotor neurons and lumber 4–5 SpMNs from P100 hSOD1 G93A mice. Motor neurons were stained by ChAT antibody in red. hSOD1 aggregates were present in SpMNs and stained by misfolded hSOD1 antibody in green. Confocal laser intensity used for imaging misfolded human SOD1 was 2% in oculomotor neurons and 0.2% in lumber SpMNs (laser intensity for ChAT imaging was same for both motor neuron types). hSOD1 staining in lumber

*Figure 1 continued on next page*

*Figure 1 continued*

SpMNs with 2% laser intensity is available in *Figure 1—figure supplement 1* (A). (B) Quantification of percentage of cells containing hSOD1 aggregates in oculomotor neurons and SpMNs from hSOD1 G93A mice (n = 3 animals, m = 40 ~ 100, mean ± SD). (C) Schematic diagram of isolation of primary motor neurons from E11.5 *Isl^MN:GFP*-positive mouse embryos. Primary motor neurons were dissected, dissociated and isolated using fluorescence-activated cell sorting (FACS). (D) Experimental outline of primary motor neuron survival assay: the number of living cells was measured on days 2, 7 and 14 in vitro (DIV). Primary O/TrMNs survived better than primary SpMNs (n = 4, mean ± SEM). (E) Differentiation scheme of ESCs to motor neurons: isogenic iNIP and iNIL ESCs were differentiated into iCrMNs and iSpMNs, respectively, by direct programming. (F) Experimental outline of ESC to motor neuron differentiation followed by survival assay. (G) iCrMNs survived better than iSpMNs (n = 7–8, mean ± SEM). (H) tSNE plot of single cell RNA sequencing data of Day 2–7 ESC-derived motor neurons and DIV seven primary motor neurons. Same plot labeled by cluster is shown in *Figure 1—figure supplement 1* (B). (I) All samples were plotted by PC2 that represents CrMN versus SpMN fates. n = biological replicates; m = number of cells quantified per replicate; statistical analysis was performed by student's t-test, *p<0.05, **p<0.01, ***p<0.001.

DOI: https://doi.org/10.7554/eLife.44423.002

The following figure supplement is available for figure 1:

**Figure supplement 1.** Comparative transcriptome analysis of ESC-derived and primary CrMNs and SpMNs by single cell RNA sequencing.

DOI: https://doi.org/10.7554/eLife.44423.003

culture and differentiation conditions for CrMNs and SpMNs to model and understand the intrinsic mechanisms that contribute to motor neurons' differential ability to resist ALS-induced stress.

Since culture conditions to maintain motor neurons were developed for SpMNs, we wanted to establish conditions that will support both SpMNs and CrMNs derived from mouse embryos and ESCs. Maintaining differentiated SpMNs in vitro requires trophic support that is normally provided by co-culture with other cells, typically glia. However, the presence of other cell types in the culture complicates the investigation of cell-autonomous neuronal properties because treatments might act on supporting cells that then affect neuronal physiology. Mixed cultures also complicate and, in some cases, preclude the use of bulk measurements such as RNA sequencing and western blot. Therefore, to study the intrinsic properties of motor neurons, we first established a mono-culture condition that supported both primary CrMNs and SpMNs in vitro.

Primary oculomotor and trochlear motor neurons (O/TrMNs) were dissected and dissociated from embryonic day 11.5 (E11.5) *Isl^MN:GFP* transgenic mice (*Lewcock et al., 2007*), isolated using fluorescence-activated cell sorting (FACS), and plated into 96-well plates (*Figure 1C*). Survival of primary O/TrMNs and SpMNs was measured on days 7 and 14 in vitro (DIV) and compared to DIV 2 (*Figure 1D*). With the optimized supplement cocktail [10 ng/mL GDNF, 10 ng/mL BDNF, 10 ng/mL CNTF, 100 µM 3-isobutyl-1-methylxanthine (IBMX) and 10 µM Forskolin], 88% of primary O/TrMNs and 72% of primary SpMNs survived on DIV 7 (*Figure 1D*). On DIV 14, 71% of primary O/TrMNs and only 42% of primary SpMNs survived. While O/TrMNs resist significantly better the stress associated with tissue dissociation, sorting and culture conditions when compared to SpMNs, both motor neuron types survive without additional cellular support.

## Rapid and efficient ESC-based platform to study differential ALS sensitivity between CrMNs and SpMNs

The O/TrMN is a small population without known unique markers that would allow the isolation of large cell quantities for biochemical studies. To overcome the difficulty of obtaining large numbers of these ALS-resistant motor neurons and to investigate possible cellular features associated with restricting SOD1 protein accumulation and enabling ALS resistance, we adopted an ESC differentiation approach. Direct programming by forced expression of transcription factors NIP (Neurogenin2-Isl1-Phox2a) and NIL (Neurogenin2-Isl1-Lhx3) (*Mazzoni et al., 2013*) efficiently generates large numbers of iCrMNs and iSpMNs from mouse ESCs respectively (*Figure 1E*).

We tested the survival of ESC-derived motor neurons using the optimized supplement cocktail described above. iCrMNs and iSpMNs were dissociated from embryoid bodies after 48 hr induction of transcription factors and plated as single cells on Day 0 (*Figure 1F*). By Day 5, 87% of iCrMNs and 74% of iSpMNs survived compared to Day 2 (*Figure 1G*). By Day 25, 52% of iCrMNs and 31% of iSpMNs survived. Thus, ESC-derived iCrMNs and iSpMNs resemble the survival difference between ALS-resistant CrMNs and ALS-sensitive SpMNs.

To address the transcriptomic similarity between ESC-derived neurons and primary neurons, we performed single-cell RNA sequencing on ESC-derived iCrMNs and iSpMNs cultured for 2, 5 and 7

days and primary O/TrMNs and SpMNs cultured for 7 days in a similar medium. Clustering of all samples by T-distributed Stochastic Neighbor Embedding (t-SNE) plot is dominated by two factors: ESC versus primary origin, and CrMN versus SpMN fate (*Figure 1H* and *Figure 1—figure supplement 1B*). While there is no co-clustering of iCrMNs and iSpMNs with embryonically-derived O/TrMNs and SpMNs, respectively, about half of ESC-derived Day 5 and 7 iCrMNs formed clusters.

Principle component analysis (PCA) suggests that PC1 separated in vitro versus in vivo differentiated neurons (*Figure 1—figure supplement 1C*) and that PC2 separated O/TrMN fate versus SpMN fate (*Figure 1I*). PC2 separated primary O/TrMNs and SpMNs. As ESC-derived motor neurons mature over days in culture, iCrMNs and iSpMNs adopted more of O/TrMN and SpMN signatures, respectively (*Figure 1I*). Single-cell RNA sequencing also confirmed that CrMN and SpMN markers were differentially enriched in ESC-derived iCrMNs and iSpMNs (*Figure 1—figure supplement 1D*). We noted that, as time progresses, some iSpMNs drifted away from primary SpMNs in the PC2 dimension (*Figure 1I*). The cellular heterogeneity might be explained by variance in differentiation and maturation speed, as well as by in vitro culture stress that constantly selects for cells with higher resilience. Thus, the comparative transcriptome analysis further supports the notion that ESC-derived iCrMNs and iSpMNs resemble molecular features of the ALS-resistant O/TrMNs and ALS-sensitive SpMNs, respectively.

Having established differentiation and culture conditions to maintain iCrMNs and iSpMNs, we next sought to test if iCrMNs and iSpMNs reacted differently to the expression of proteins associated with familial ALS. Inspired by the finding that over-expression of mutant hSOD1 induces ALS-like symptoms in mouse (*Gurney et al., 1994*; *Wong et al., 1995*), we generated mouse ESC lines expressing high levels of wild type (WT) hSOD1, and hSOD1 with A4V or G93A mutations, both of which are genetic causes of familial ALS (*Rosen, 1993*; *Gurney et al., 1994*; *Kiskinis et al., 2014*). Clones with similar expression levels of the three hSOD1 variants were selected. From these transgenic clones, we derived isogenic inducible-NIP and -NIL (iNIP and iNIL) ESC lines (*Mazzoni et al., 2013*), which were then differentiated into iCrMN and iSpMNs respectively (*Figure 2A,B*). iNIP and iNIL ESCs expressed hSOD1 variants at very similar levels (*Figure 2C,D* and *Figure 2—figure supplement 1A*). Day 2 iCrMNs and iSpMNs also contained comparable amounts of hSOD1 (*Figure 2E, F* and *Figure 1—figure supplement 1BC*). Notably, hSOD1 transgenic expression did not affect neuronal differentiation efficiency and the hSOD1 transgenes were not silenced in iCrMNs or iSpMNs after differentiation (*Figure 2G–I*).

In summary, we established a rapid and scalable motor neuron differentiation and culture platform that allowed mono-culture of both primary and ESC-derived CrMNs and SpMNs in vitro with excellent short-term survival. iCrMNs and iSpMNs recapitulated the differential survival of primary motor neurons under the same culture conditions, suggesting that ESC-derived motor neurons have the potential to represent intrinsic properties of their counterparts in vivo.

## iCrMNs accumulate less hSOD1 mutant proteins over time than iSpMNs

We next investigated whether ESC-derived iCrMNs and iSpMNs would accumulate hSOD1 in vitro by monitoring hSOD1 protein accumulation on Day two and Day 15 by western blot (*Figure 3A*). Soluble hSOD1 can be extracted by lysing the cells with a detergent-based lysis buffer (RIPA), while misfolded and aggregated hSOD1 tend to remain insoluble. However, insoluble hSOD1 can be solubilized from the remaining precipitate using urea (*Kiskinis et al., 2014*). As a result, the RIPA fraction is enriched with soluble hSOD1 while the urea fraction is enriched with insoluble hSOD1 aggregates. Mirroring what has been observed in ALS patients and mice with hSOD1 mutations (*Da Cruz et al., 2017*), over-expressed hSOD1 mutant proteins accumulated in ESC-derived iSpMNs over time (*Figure 3B,C*). While iCrMNs maintained similar levels of soluble hSOD1 mutant proteins from Day two to Day 15, iSpMNs accumulated over two times more soluble hSOD1 A4V and G93A by Day 15 compared to Day 2 (*Figure 3B,C*). Moreover, iSpMNs also accumulated ~3 times more insoluble hSOD1 A4V and G93A than iCrMNs on Day 15 (*Figure 3B,D*). Thus, iCrMNs accumulated less mutant hSOD1 than iSpMNs over time (*Figure 3B,D*).

The differential accumulation of hSOD1 proteins between iCrMNs and iSpMNs was not secondary to cell-specific silencing or upregulation of the transgene as both the hSOD1 A4V and hSOD1 G93A transgenes were expressed at similar levels between iCrMNs and iSpMNs on Day 7, measured by RT-qPCR. The relative expression of hSOD1 A4V compared to beta-actin was $1.17 \pm 0.66$ in iCrMNs and $1.08 \pm 0.38$ in iSpMNs (iCrMNs versus iSpMNs p=0.84 by Student's t-test, n = 3) and the relative

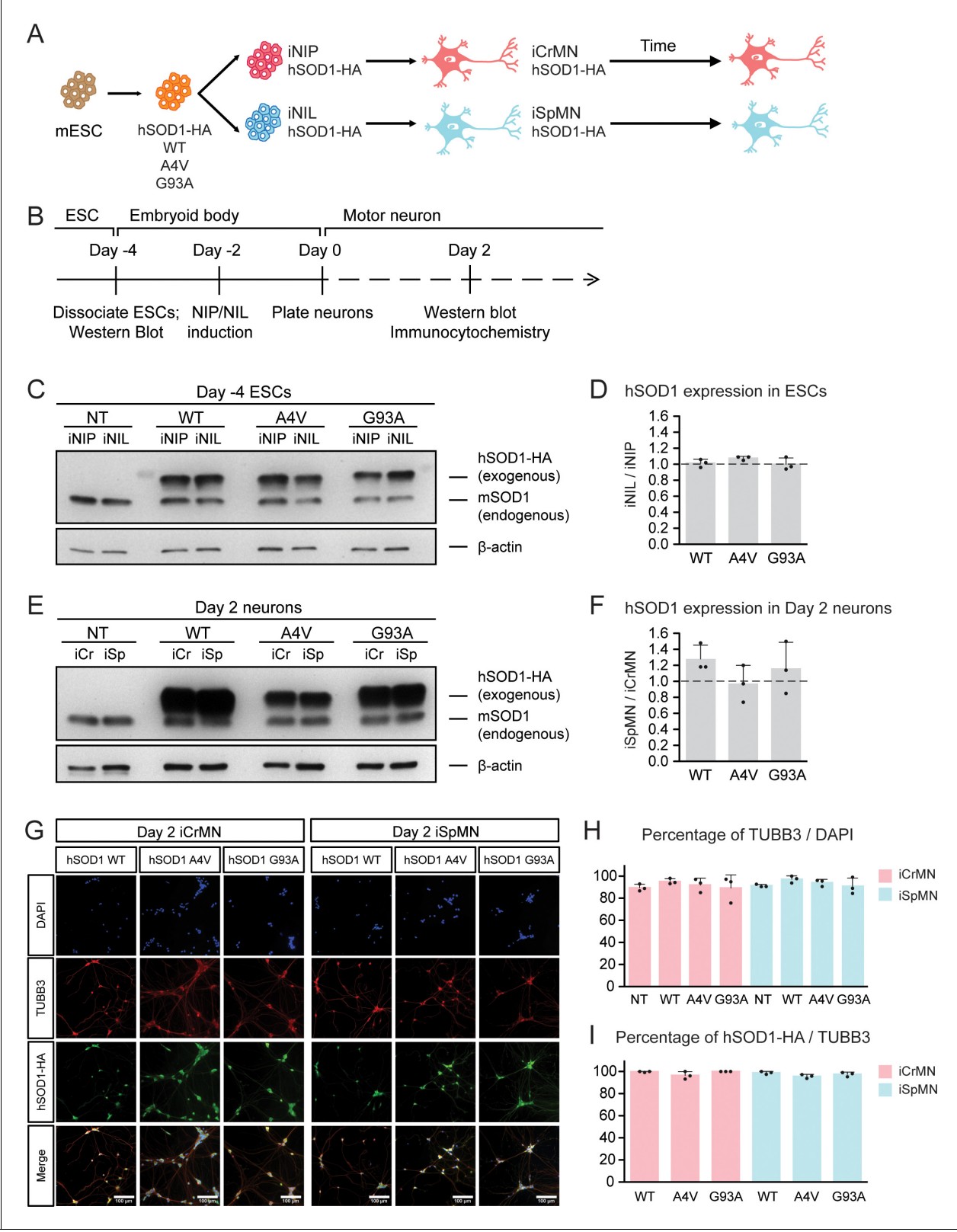

**Figure 2.** Generation of ESC-derived iCrMNs and iSpMNs expressing similar level of wild type or ALS mutant hSOD1 proteins. (**A**) Schematic diagram of generation of ESC-derived motor neurons expressing hSOD1 transgenes: transgenic ESC lines expressing wild type (WT), A4V or G93A hSOD1 were first constructed; from these, isogenic iNIP and iNIL ESC lines were derived and then differentiated into motor neurons by direct programming. (**B**) Experimental outline: motor neurons were differentiated and quantified by immunocytochemistry on Day 2. Expression of hSOD1 was quantified by

*Figure 2 continued on next page*

*Figure 2 continued*

western blot in ESCs and Day two neurons. (C) Western blot analysis of ESCs using SOD1 antibody that recognizes both endogenous mouse SOD1 (mSOD1) and exogenous hSOD1 with HA tag. (D) Quantification of the hSOD1 expression between iNIL and iNIP ESCs (n = 3, mean ± SD). (E) Western blot analysis of Day two neurons using SOD1 antibody. iCr: iCrMN; iSp: iSpMN. (F) Quantification of the hSOD1 expression between iSpMNs and iCrMNs (n = 3, mean ± SD). Quantifications of the fold change between exogenous hSOD1 versus endogenous mSOD1 expression in ESCs and neurons are available in *Figure 2—figure supplement 1*. (G) Representative images of Day two neurons for quantification. All cells were stained by DAPI and neurons were stained by pan-neuronal marker TUBB3. hSOD1 proteins were stained by HA antibody. (H) Quantification of percentage of TUBB3$^+$ cells on Day 2 (n = 3, m =~ 100, mean ± SD). (I) Quantification of percentage of hSOD1-HA$^+$ cells in TUBB3$^+$ neurons on Day 2 (n = 3, m =~ 100, mean ± SD). n = biological replicates; m = number of cells quantified per replicate. β-actin was used as a loading control for normalization in (D) and (F). The dash line represents y = 1. Statistical analysis was performed in log-transformed data by student's t-test, *p<0.05, **p<0.01, ***p<0.001.
DOI: https://doi.org/10.7554/eLife.44423.004

The following figure supplement is available for figure 2:

**Figure supplement 1.** Overexpression of transgenic hSOD1 variants.
DOI: https://doi.org/10.7554/eLife.44423.005

expression of hSOD1 G93A compared to beta-actin was 3.29 ± 0.44 in iCrMNs and 3.51 ± 0.56 in iSpMNs (iCrMNs versus iSpMNs p=0.64 by Student's t-test, n = 3). The translation efficiency of hSOD1 was also similar between iCrMNs and iSpMNs as measured by polysome profiling (*Figure 3— figure supplement 1*). Therefore, it is likely that iCrMNs have a cell-autonomous mechanism to restrain the accumulation of hSOD1 mutant proteins, and perhaps other misfolded proteins.

## Proteasomal degradation reduces mutant hSOD1 accumulation in iCrMNs

The ubiquitin-proteasome system is the prime mechanism to degrade soluble proteins, and involves two discrete and successive steps: covalent modification of the target protein with ubiquitin and subsequent degradation of the tagged protein in the catalytic 20S core of the 26S proteasome (*Dantuma and Bott, 2014*). To investigate if iCrMNs and iSpMNs use the proteasome pathway differentially to degrade hSOD1 mutant proteins, we compromised the proteasome pathway and measured the amount of hSOD1 accumulation. To avoid pleiotropic toxic effects, we applied a milder treatment of proteasome inhibitor MG-132 than used in previous reports that tested the proteasomal degradation of SOD1 and TDP-43 (*Kiskinis et al., 2014*; *Scotter et al., 2014*). Specifically, we treated Day 4 hSOD1 expressing neurons with DMSO (vehicle control) or 100 nM MG-132 for 18 hr. 100 nM of MG-132 reduced proteasome activity by ~30% in both iCrMNs (68.27 ± 10.30% compared to DMSO) and iSpMNs (67.63 ± 9.49% compared to DMSO, iCrMNs versus iSpMNs p=0.47 by Student's t-test, n = 3). The proteasome activity was measured as the hydrolysis rate of N-succinyl-Leu-Leu-Val-Tyr-7-amido-4-methylcoumarin (Suc-LLVY-AMC) (*Meng et al., 1999*).

We then compared soluble and insoluble hSOD1 accumulation in the DMSO and MG-132 treated neurons by western blot (*Figure 4A*). The ~30% reduction of proteasome activity did not significantly affect levels of soluble WT or mutant hSOD1 protein in iCrMNs or iSpMNs (*Figure 4B,C*). By contrast, the accumulation of insoluble hSOD1 A4V and G93A mutant proteins (but not WT) was significantly affected by MG-132 treatment (*Figure 4B,C*). iSpMNs accumulated ~2 fold more insoluble mutant hSOD1 when the proteasome was compromised (1.73 ± 0.21 fold more hSOD1 A4V, p=0.03; 1.93 ± 0.23 fold more hSOD1 G93A, p=0.02) (*Figure 4C*), suggesting that proteins that would have normally been degraded by the proteasome instead contributed to the insoluble aggregates. Revealingly, iCrMNs accumulated ~8.5 fold more insoluble hSOD1 A4V and ~5.3 fold more insoluble hSOD1 G93A compared to DMSO control (*Figure 4B,C*). Thus, a substantial fraction of mutant hSOD1 protein is being targeted for degradation by the proteasome in iCrMNs. Moreover, overexpressed hSOD1 A4V and G93A are more prone to misfolding than overexpressed WT hSOD1, which was presented by a reduction in proteasome activity. These results imply that iCrMNs may rely on the proteasome to a greater extent than iSpMNs to degrade mutant misfolded hSOD1 proteins and reduce their accumulation in insoluble aggregates.

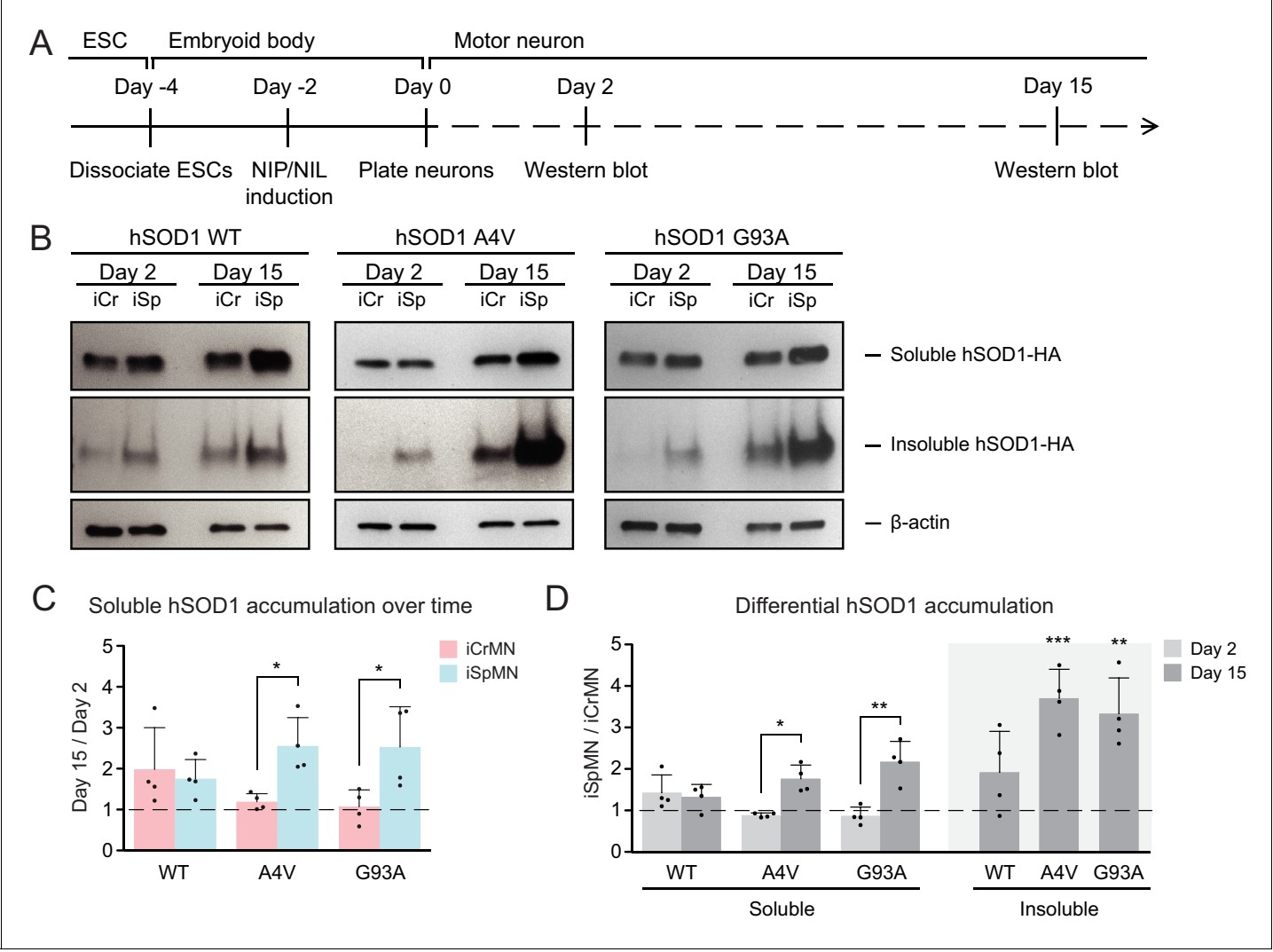

**Figure 3.** iCrMNs accumulate less mutant hSOD1 proteins than iSpMNs over time. (**A**) Experimental outline: motor neurons were differentiated and the level of hSOD1 was quantified by western blot on Day two and Day 15. (**B**) Western blot analysis of Day two and Day 15 iCrMNs and iSpMNs using HA antibody. Accumulation of hSOD1 proteins is shown in the soluble (RIPA) and insoluble (urea) fractions. iCr: iCrMN; iSp: iSpMN. (**C**) Quantification of soluble hSOD1 level between Day 15 and Day two neurons (n = 4, mean ± SD). (**D**) Quantification of hSOD1 accumulation between iSpMNs and iCrMNs on Day two and Day 15 (n = 4, mean ± SD). β-actin in the soluble fraction was used as a loading control for normalization in (**C**) and (**D**). The dash line represents y = 1. n = biological replicates; statistical analysis was performed in log-transformed data by student's t-test, *p<0.05, **p<0.01, ***p<0.001.

DOI: https://doi.org/10.7554/eLife.44423.006

The following figure supplement is available for figure 3:

**Figure supplement 1.** The translation efficiency of hSOD1 is similar between iCrMNs and iSpMNs.

DOI: https://doi.org/10.7554/eLife.44423.007

## iCrMNs rely on proteasome function to reduce the accumulation of insoluble protein aggregates

CrMN resistance to ALS neurodegeneration is not limited to SOD1-linked ALS. It is conceivable that iCrMNs rely more on the proteasome pathway to prevent the accumulation not just of hSOD1, but of insoluble proteins in general. Proteins to be degraded by the proteasome are tagged by ubiquitin, so we next investigated whether reducing proteasome activity would result in widespread accumulation of ubiquitinated proteins in addition to accumulation of mutant hSOD1. To this end, we again treated iCrMNs and iSpMNs with MG-132 and DMSO, but probed the western blot with an

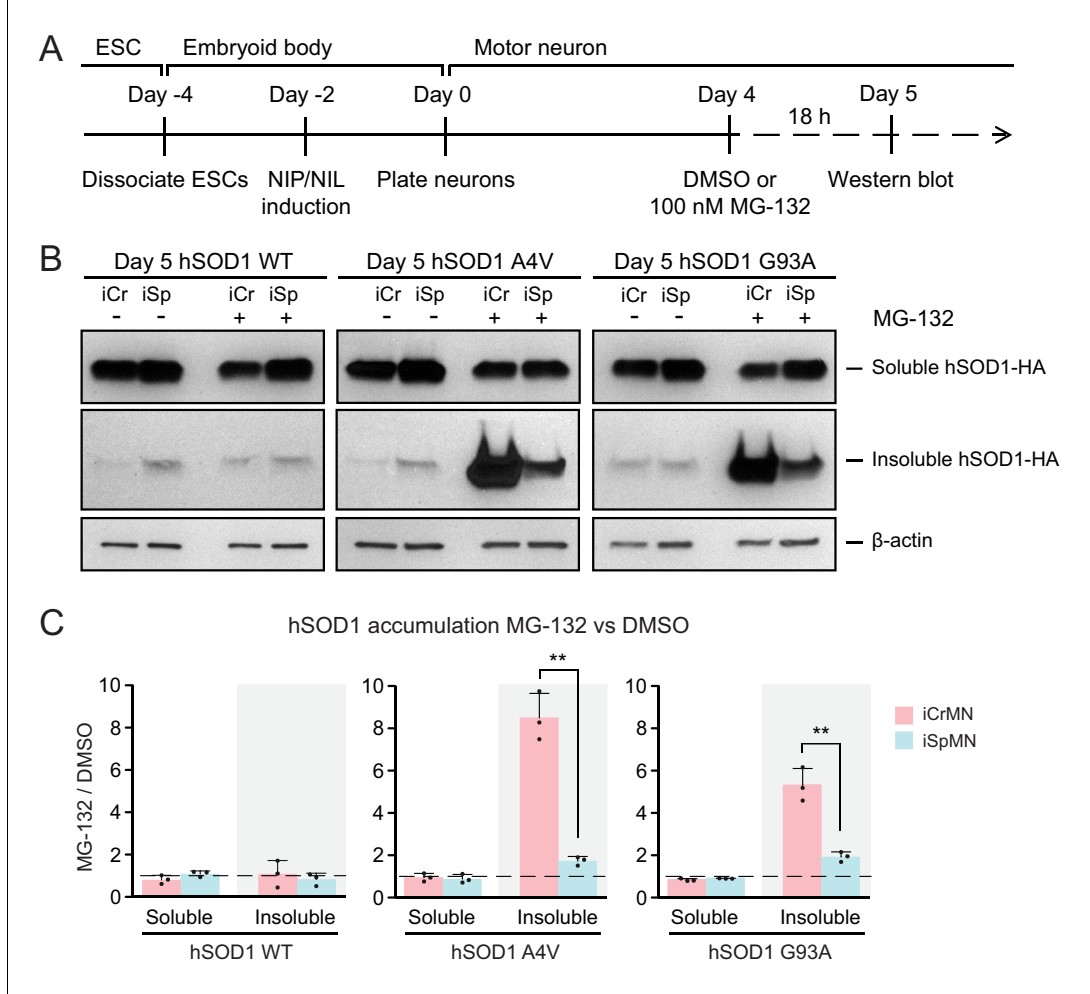

**Figure 4.** Proteasome dependent degradation reduces insoluble mutant hSOD1 accumulation in iCrMNs. (**A**) Experimental outline: motor neurons were differentiated and treated with DMSO or 100 nM MG-132 for 18 hr on Day four followed by western blot analysis on Day 5. (**B**) Western blot analysis of Day 5 DMSO or MG-132 treated iCrMNs and iSpMNs using HA antibody. Accumulation of hSOD1 proteins is shown in the soluble (RIPA) and insoluble (urea) fractions. iCr, iCrMN; iSp, iSpMN. (**C**) Quantification of hSOD1 accumulation between MG-132 and DMSO treated neurons (n = 3, mean ± SD). β-actin in the soluble fraction was used as a loading control for normalization. The dashed line represents equal accumulation, y = 1. n = biological replicates; statistical analysis was performed in log-transformed data by Two-way ANOVA and student's t-test, *p<0.05, **p<0.01, ***p<0.001.
DOI: https://doi.org/10.7554/eLife.44423.008

anti-ubiquitin antibody (*Figure 5A*). Following MG-132 treatment, there was a visible increase in ubiquitinated proteins in both iCrMNs and iSpMNs that appeared as a smear of bands in both the soluble and insoluble fractions (*Figure 5B*). Mirroring the response of hSOD1 mutant proteins, there was significantly greater accumulation of insoluble ubiquitinated proteins in MG-132 treated iCrMNs compared to iSpMNs (*Figure 5C*). This response to MG-132 treatment was universal across WT, A4V and G93A hSOD1 expressing neurons (13–18 fold and 3–5 fold more ubiquitinated proteins in iCrMNs and iSpMNs respectively). Therefore, we conclude that there is an intrinsic difference between iCrMNs and iSpMNs in their use of the ubiquitin proteasome system to degrade misfolded proteins that would otherwise contribute to insoluble protein aggregates. p62 is a ubiquitin-binding protein and a common component of ubiquitinated inclusions in degenerating SpMNs of ALS patients and mouse models (*Seibenhener et al., 2004*; *Liu et al., 2016*; *Mizuno et al., 2006*; *Gal et al., 2007*). We investigated if there was concomitant accumulation of p62 with the dramatic increase of insoluble ubiquitinated proteins in MG-132 treated iCrMNs. The mild reduction of proteasome activity resulted in accumulation of more soluble and insoluble p62 proteins in both iCrMNs and iSpMNs, with iSpMNs and iCrMNs accumulating 2–3 fold and 6–10 fold more insoluble p62

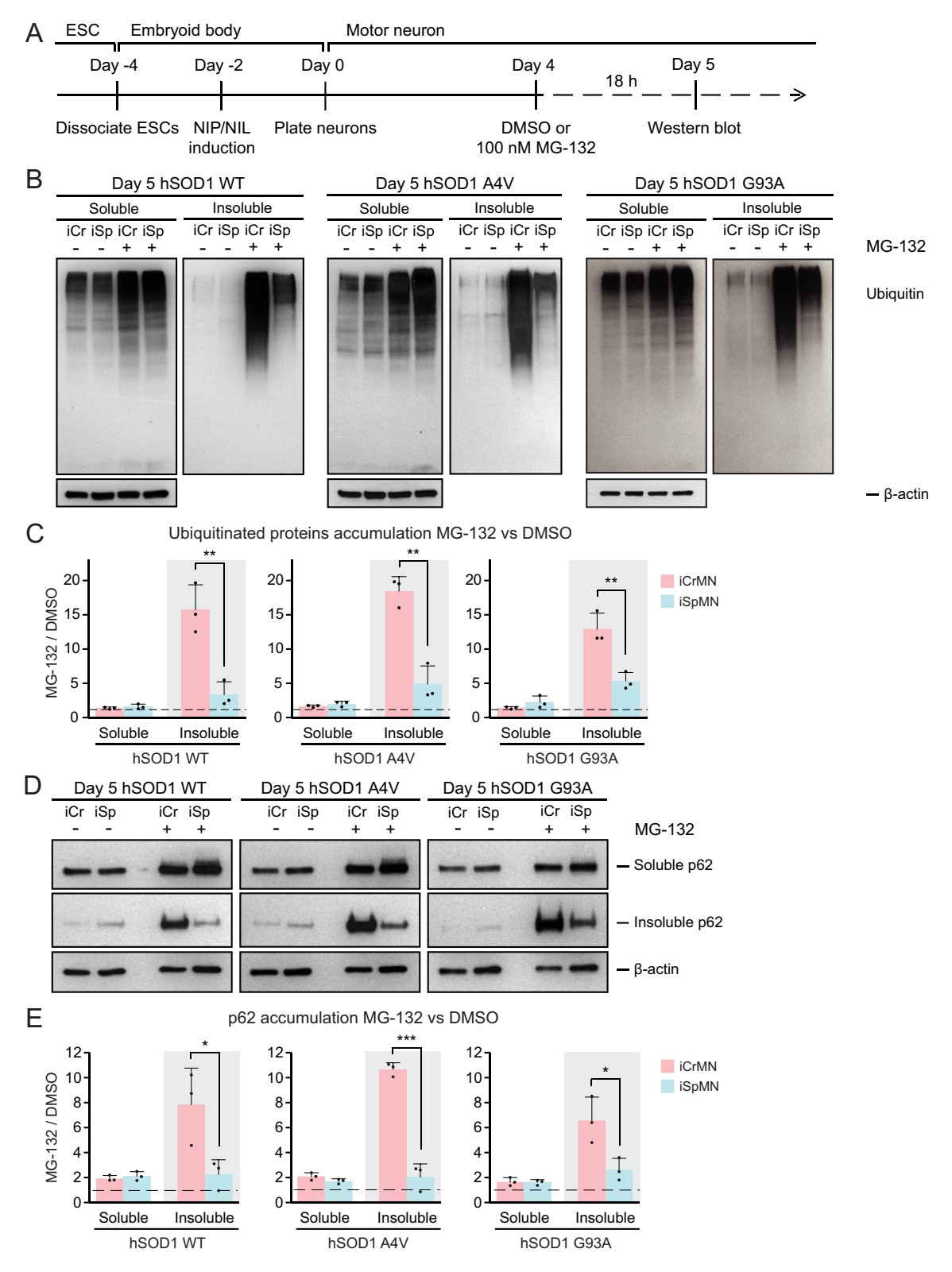

**Figure 5.** iCrMNs rely on proteasome function to reduce the accumulation of insoluble protein aggregates. (**A**) Experimental outline: motor neurons were differentiated and treated with DMSO or 100 nM MG-132 for 18 hr on Day four followed by western blot analysis on Day 5. (**B**) Western blot analysis of Day 5 DMSO or MG-132 treated iCrMNs and iSpMNs using ubiquitin antibody revealed a smear of ubiquitinated proteins in the soluble (RIPA) and insoluble (urea) fractions. iCr, iCrMN; iSp, iSpMN. (**C**) Quantification of accumulation of ubiquitinated proteins between MG-132 and DMSO

*Figure 5 continued on next page*

*Figure 5 continued*

treated neurons (n = 3, mean ± SD). (**D**) Western blot analysis of Day 5 DMSO or MG-132 treated iCrMNs and iSpMNs using p62 antibody. Accumulation of p62 proteins is shown in soluble (RIPA) and insoluble (urea) fractions. iCr, iCrMN; iSp, iSpMN. (**E**) Quantification of p62 accumulation between MG-132 and DMSO treated neurons (n = 3, mean ± SD). β-actin in the soluble fraction was used as a loading control for normalization in (**C**) and (**E**). The dashed line represents equal accumulation, y = 1. n = biological replicates; statistical analysis was performed in log-transformed data by Two-way ANOVA and student's t-test, *p<0.05, **p<0.01, ***p<0.001.
DOI: https://doi.org/10.7554/eLife.44423.009

protein respectively (*Figure 5D,E*). Therefore, similar to the response behavior of ubiquitinated proteins, iCrMNs accumulated significantly more insoluble p62 proteins than iSpMNs after MG-132 treatment (*Figure 5D,E*).

Overall, a 30% reduction of proteasome activity induced greater accumulation in iCrMNs than in iSpMNs of insoluble hSOD1 mutant proteins, of insoluble ubiquitinated proteins, and of the insoluble ubiquitin-binding protein p62. These results suggest that iCrMNs are, in general, more efficient at proteasomal degradation of misfolded proteins than iSpMNs.

## ESC- and embryonically-derived ALS resistant CrMNs contain fewer p62 protein aggregates than SpMNs

Efficient proteasomal degradation of misfolded proteins in iCrMNs suggests a higher capability of maintaining protein homeostasis. Thus, we hypothesized that iCrMNs would accumulate less p62 in the insoluble fraction than iSpMNs over time. To test this hypothesis, we quantified the soluble and insoluble p62 levels in Day 15 iCrMNs and iSpMNs by western blot (*Figure 6A*). While both iCrMNs and iSpMNs accumulate p62 in insoluble aggregates, iSpMNs expressing mutant hSOD1 accumulated significantly more insoluble p62 than iCrMNs (*Figure 6B,C*). Since p62 aggregation is an indicator of ubiquitinated inclusions, this result additionally supports the concept that iCrMNs are better at maintaining protein homeostasis in culture.

The differential p62 accumulation in ESC-derived motor neurons suggests that ALS-resistant O/TrMNs would have fewer p62-containing inclusions than SpMNs in an ALS mouse model. To test this hypothesis, we measured the percentage of motor neurons with p62 positive protein aggregates in the hSOD1 G93A mouse model at postnatal days 66 and 97 (*Gurney et al., 1994*; *Watanabe et al., 2001*). While over 40% of lumbar SpMNs accumulated p62 positive inclusions, there were no oculomotor neurons with protein aggregates containing p62 (*Figure 6D,E*). Thus, as predicted by the iCrMNs and iSpMNs programmed in vitro, the ALS-resistant CrMNs have the capacity to prevent the accumulation of p62-containing inclusions.

We then asked if the differential accumulation of p62-containing protein aggregates, a typical cytopathological feature of ALS patients and mouse models, could be explained by a differential use of the proteasome degradation pathway similar to what we observed by western blot. To that end, we quantified p62 positive cytoplasmic protein aggregates via immunocytochemistry in DMSO or MG-132 treated iCrMNs and iSpMNs expressing mutant hSOD1 on Day 5 (*Figure 6A*). Consistent with in vivo observations, more iSpMNs (~15%) exhibited p62-positive cytoplasmic aggregates than did iCrMNs (~5%) (*Figure 6F*). Interestingly, inhibition of the proteasome by MG-132 treatment had a greater effect on iCrMNs, nearly 50% of which contained p62 aggregates, compared to ~30% of iSpMNs (*Figure 6F*). These data suggest that CrMNs use the proteasome pathway more efficiently than SpMNs to prevent the accumulation of p62-containing insoluble aggregates.

## iCrMNs are more resistant than iSpMNs to proteostatic stress

During ALS progression, SpMNs are under proteostatic stress that triggers the unfolded protein response (UPR) (*Atkin et al., 2008*; *Matus et al., 2013*; *Hetz and Mollereau, 2014*). Our results suggest that iCrMNs are intrinsically better than iSpMNs at coping with misfolded proteins and, thus, at maintaining a healthier proteome. We hypothesized that under generalized proteostatic stress, iCrMN's ability to better maintain proteostasis would result in increased survival when compared to iSpMNs. To test if iCrMNs are intrinsically more resistant than iSpMNs to proteostatic stress, we compared the survival of non-transgenic iCrMNs and iSpMNs when treated with two well-described proteostatic stressors, tunicamycin and cyclopiazonic acid (CPA) (*Merlie et al., 1982*;

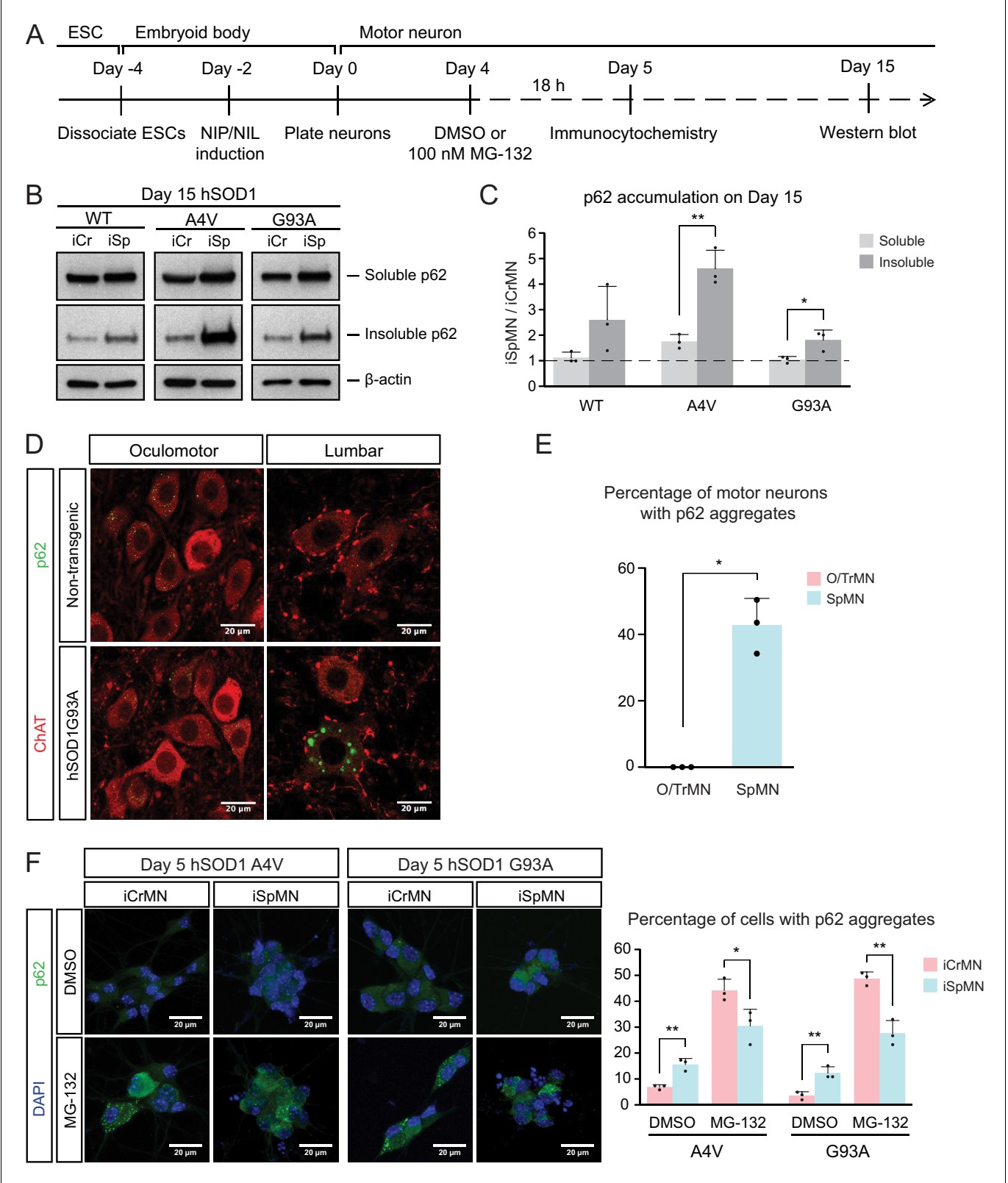

**Figure 6.** ESC- and embryonically-derived ALS-resistant CrMNs contain fewer p62-positive protein aggregates than SpMNs. (**A**) Experimental outline: motor neurons were differentiated and treated with DMSO or 100 nM MG-132 for 18 hr on Day four followed by immunocytochemistry analysis on Day 5; neurons were collected on Day 15 for western blot analysis. (**B**) Western blot analysis of Day 15 neurons using p62 antibody. Accumulation of p62 proteins is shown in the soluble (RIPA) and insoluble (urea) fractions. iCr, iCrMN; iSp, iSpMN. (**C**) Quantification of p62 levels between Day 15 iCrMNs

*Figure 6 continued on next page*

*Figure 6 continued*

and iSpMNs (n = 3, mean ± SD). β-actin in the soluble fraction was used as a loading control for normalization. The dashed line represents y = 1. (D) Representative images of oculomotor neurons and lumber 4–5 SpMNs from littermates of non-transgenic (NT) and hSOD1 G93A mice at postnatal days 66, 66 and 97. ChAT antibody stained motor neurons in red. p62 antibody stained p62-containing inclusions in green. (E) Quantification of percentage of cells containing p62 inclusions in oculomotor neurons and SpMNs from hSOD1 G93A mice (n = 3, m =~ 100, mean ± SD). (F) Representative images of Day five neurons treated with DMSO or MG-132 stained by DAPI in blue and p62 antibody in green. Quantification of percentage of cells that contain p62 aggregates is shown on the right (n = 3, m =~ 100, mean ± SD). n = biological replicates or animals; m = number of cells quantified per replicate; statistical analysis was performed in log-transformed data by Two-way ANOVA and student's t-test, *p<0.05, **p<0.01, ***p<0.001.
DOI: https://doi.org/10.7554/eLife.44423.010

*Seidler et al., 1989*) (*Figure 7A*). Tunicamycin and CPA induce misfolded proteins by different mechanisms. Tunicamycin inhibits N-linked glycosylation and thus affects trans-membrane and secreted proteins. CPA blocks sarco/endoplasmic reticulum $Ca^{2+}$-ATPase (SERCA), the $Ca^{2+}$-pump in endoplasmic reticulum (ER) membrane that helps to maintain $Ca^{2+}$ balance and ER homeostasis, and thus has a broader effect. Notably, CPA was reported to accelerate the degeneration of SpMNs expressing hSOD1 G93A mutant, suggesting that CPA treatment may reveal ALS-relevant differential vulnerability between iCrMNs and iSpMNs (*Thams et al., 2019*).

Survival experiments revealed that iCrMNs are more resistant to tunicamycin than iSpMNs. While 100 ng/ml tunicamycin affected both cell types nearly equally (88 ± 4% survival for iCrMNs and 83 ± 5% for iSpMNs, respectively), 150 ng/ml tunicamycin was significantly more toxic to iSpMNs than to iCrMNs (65 ± 6% survival for iCrMNs and 47 ± 3% for iSpMNs, respectively) (*Figure 7B*). iCrMNs were consistently more resistant to tunicamycin up through the highest concentration tested (300 ng/mL), at which close to 50% (49% ± 5%) of iCrMNs survived compared to only about 10% (11% ± 1%) of iSpMNs.

iCrMNs are also more resistant to CPA than iSpMNs. Both iCrMNs and iSpMNs were largely unaffected by 5 μM CPA. 10 μM CPA, however, was sufficient to reveal a survival difference, with iCrMNs displaying an insensitivity to treatment (103 ± 6%) compared to only ~80% (75% ± 4%) of iSpMNs surviving (*Figure 7C*). This differential survival trend was maintained through the highest concentration tested (40 μM), at which the survival of iSpMNs was only about 10% (9% ± 1%) compared to nearly 40% (37% ± 3%) of iCrMNs. To exclude drug specific effects, we also conducted a survival assay with thapsigargin, which has the same target as CPA, and found results consistent with CPA treatment (*Figure 7D*). To compare the molecular response of iCrMN and iSpMN to chemically induced proteostatic stress, we measured the mRNA levels of the UPR markers sliced-XBP1 (sXBP1), GRP78, ATF4 and HRD1 after 12 hr of 20 μM CPA or DMSO treatment by RT-qPCR. iSpMNs showed higher induction of sXBP1, GRP78 and ATF4 than iCrMNs (*Figure 7—figure supplement 1*). This suggests that iSpMNs experienced a stronger level of proteostatic stress than iCrMNs to the same CPA treatment, presumably because they are less efficient than iCrMNs at degrading misfolded proteins.

To test if the intrinsic ability of motor neurons to cope with misfolded hSOD1 is affected by a secondary stress, we treated the hSOD1 A4V expressing iCrMNs and iSpMNs with DMSO or 20 μM CPA and quantified the accumulation of hSOD1 A4V by western blot (*Figure 7—figure supplement 2A*). CPA treatment significantly increased the accumulation of insoluble hSOD1 A4V (*Figure 7—figure supplement 2B*). iCrMNs accumulated less insoluble hSOD1 A4V than iSpMNs regardless of the treatment (*Figure 7—figure supplement 2C*). These results suggest that additional proteostatic stress accelerates the accumulation of insoluble hSOD1 mutant proteins and that iCrMNs maintain a superior capability to deal with misfolded proteins under CPA treatment.

To further investigate iCrMNs' and iSpMNs' sensitivity to different kinds of proteostatic stress, we applied brefeldin A. Brefeldin A disrupts proteostasis by blocking intracellular protein transport from ER to Golgi, resulting in organelle failure. Brefeldin A was equally toxic to both cell types (*Figure 7E*). The comparable sensitivities of iCrMNs and iSpMNs to brefeldin A suggest that culture conditions do not favor iCrMNs' ability to resist all chemical stress.

**Primary ALS-resistant CrMNs are more resistant than primary SpMNs to proteostatic stress**
iCrMNs are more resistant to proteostatic stress than iSpMNs, suggesting that this could be an intrinsic mechanism by which embryonically derived CrMN nuclei resist ALS. To test if ALS-resistant neuronal fate is associated with an intrinsically superior capacity to deal with generalized

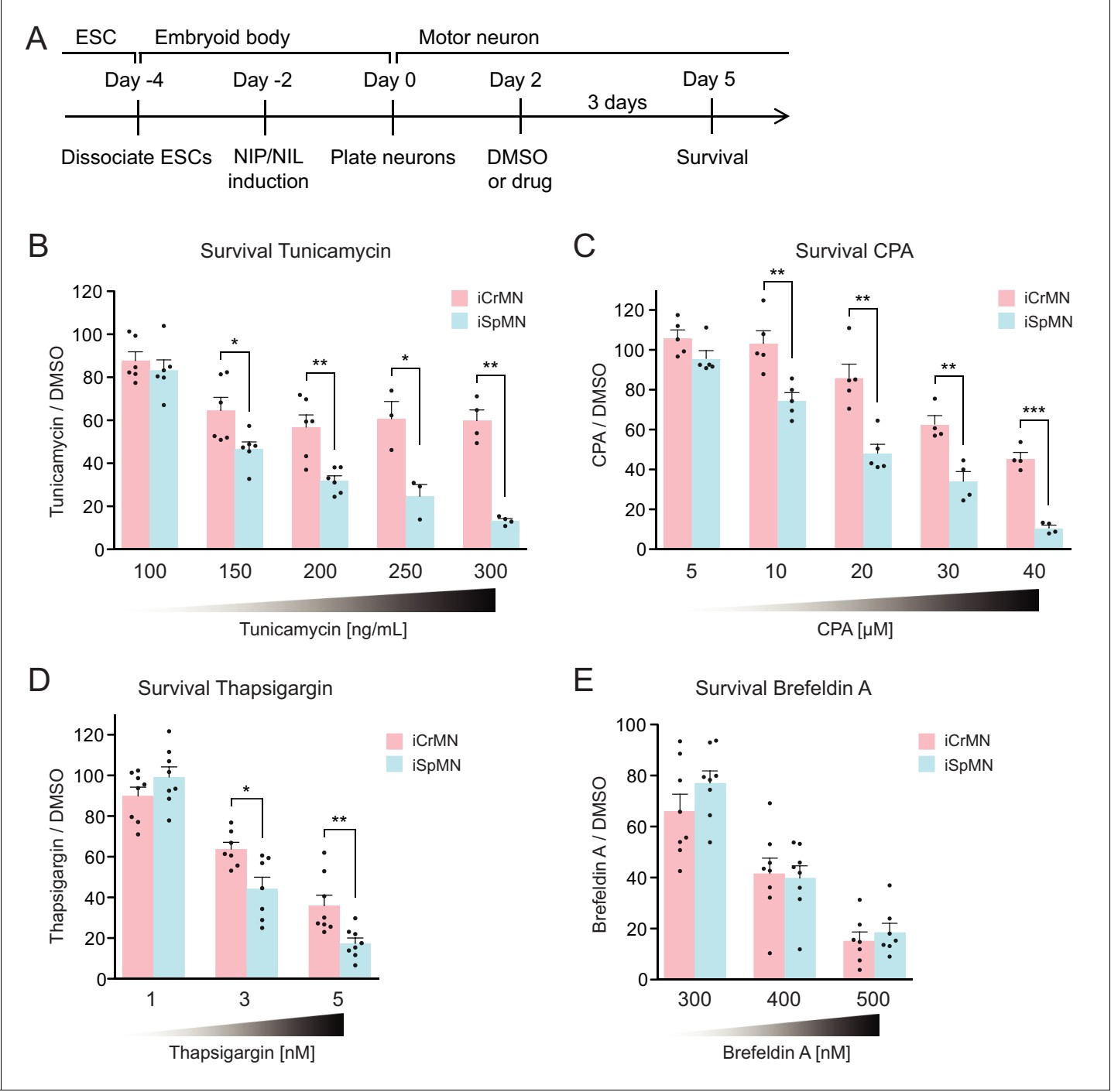

**Figure 7.** iCrMNs are more resistant than iSpMNs to proteostatic stress caused by misfolded proteins. (**A**) Experimental outline: motor neurons were differentiated and treated with DMSO or drugs from Day two to Day five and the number of living cells were measured on Day five for survival assay. (**B**) iCrMNs were more resistant than iSpMNs to tunicamycin treatment (n = 3–6, mean ± SEM). (**C**) iCrMNs were more resistant than iSpMNs to CPA treatment (n = 4–5, mean ± SEM). The UPR response induced by CPA treatment is shown in *Figure 7—figure supplement 1*. (**D**) iCrMNs were more resistant than iSpMNs to thapsigargin treatment (n = 7–8, mean ± SEM). (**E**) iCrMNs and iSpMNs were equally sensitive to brefeldin A treatment (n = 7–8, mean ± SEM). n = biological replicates; statistical analysis was performed by student's t-test, *p<0.05, **p<0.01, ***p<0.001.
DOI: https://doi.org/10.7554/eLife.44423.011

The following figure supplements are available for figure 7:

**Figure supplement 1.** CPA treatment induced higher UPR level in iSpMNs than iCrMNs Non-transgenic iCrMNs and iSpMNs were treated with 20 µM CPA on Day 2; induction of UPR markers were quantified by RT-qPCR after 12 hr.

*Figure 7 continued on next page*

*Figure 7 continued*

DOI: https://doi.org/10.7554/eLife.44423.012

**Figure supplement 2.** Additional proteostatic stress induced by CPA treatment significantly increased insoluble hSOD1 A4V accumulation in both iCrMNs and iSpMNs hSOD1 A4V expressing iCrMNs and iSpMNs were treated with 20 µM CPA from Day two to Day 5; accumulation of hSOD1 A4V proteins were quantified by western on Day 5.

DOI: https://doi.org/10.7554/eLife.44423.013

proteostatic stress, we compared the sensitivity of non-transgenic mouse primary ALS-resistant O/TrMNs and ALS-sensitive SpMNs to tunicamycin and CPA. Neurons were isolated by FACS as described above and treated with increasing concentrations of tunicamycin (50–300 ng/ml), CPA (5–30 µM) or DMSO on DIV 2 and fixed 3 days later for immunocytochemistry to evaluate survival (*Figure 8A*). Primary O/TrMNs were significantly more resistant than primary SpMNs to tunicamycin treatment within 100–250 ng/ml and to CPA treatment within 10–25 µM (*Figure 8B–C*). Taken together, primary O/TrMNs were more resistant than SpMNs to proteostatic stress caused by mis-folded proteins.

Combining the results from hSOD1 G93A mice and primary motor neuron cultures, these data confirm our hypothesis generated from ESC-derived motor neurons that ALS-resistant CrMNs are more resistant to proteostatic stress than SpMNs. It also highlights the predictive value of this ESC-based in vitro platform for the study of cell-autonomous properties that contribute to the cell specific differential vulnerability to ALS. Moreover, these in vitro and in vivo studies jointly support an intrinsic difference between ALS-resistant CrMNs and ALS-vulnerable SpMNs, one that facilitates

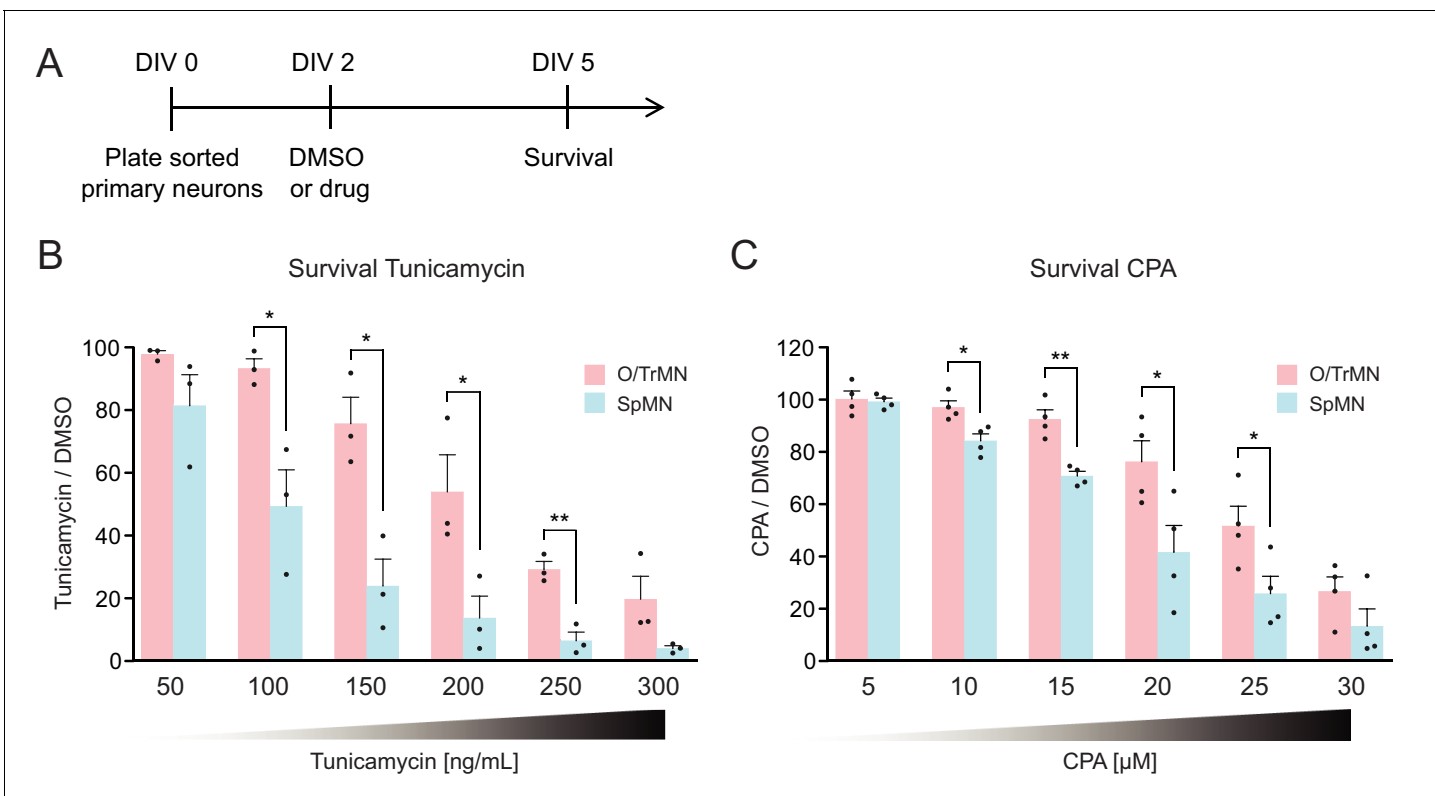

**Figure 8.** Primary ALS-resistant CrMNs are more resistant than primary SpMNs to proteostatic stress. (A) Experimental outline: primary motor neurons were dissected, dissociated, sorted and plated on DIV 0 and treated with DMSO or drugs from DIV 2 to DIV 5. The number of living cells were measured on DIV five for survival assay. (B) Primary O/TrMNs were more resistant than primary SpMNs to tunicamycin treatment (n = 3, mean ± SEM). (C) Primary O/TrMNs were more resistant than primary SpMNs to CPA treatment (n = 4, mean ± SEM). n = biological replicates; statistical analysis was performed by student's t-test, *p<0.05, **p<0.01, ***p<0.001.

DOI: https://doi.org/10.7554/eLife.44423.014

CrMNs to degrade misfolded proteins more efficiently than SpMNs. This ability allows CrMNs to accumulate fewer protein aggregates and to better resist ALS neurodegeneration.

## iCrMNs contain more proteasome 20S core subunit proteins and have a higher proteasome activity than iSpMNs

To investigate the possible molecular underpinnings behind the superior ability of iCrMNs to utilize the proteasome pathway to degrade misfolding proteins, we performed an unbiased proteomic study on Day two non-transgenic iCrMNs and iSpMNs (*Figure 9A*). Among the differences, we noted that iCrMNs contain higher levels of all 20S core subunits of the proteasome than iSpMNs (*Figure 9B*). Interestingly, these differences are not found among regulatory subunits of the proteasome (*Figure 9B*). Of note, RNA-seq experiments did not reveal this difference, which seems to be due to post-transcriptional mechanisms. While this result points to a possible mechanism, the upregulation of core proteasome proteins may not be enough to confer iCrMNs with higher proteasome activity. Thus, we sought to directly measure proteasome activity in intact iSpMNs versus iCrMNs. iCrMNs and iSpMNs were treated with Suc-LLVY-AMC that fluoresces when cleaved by the chymotrypsin-like activity of the 20S proteasome. The proteasome activity was represented by the hydrolysis rate of Suc-LLVY-AMC. Non-transgenic iCrMNs had a higher proteasome activity than iSpMNs at Day 2 and 5 in culture (*Figure 9C*). Together, these results suggest that iCrMN have a higher capacity to degrade misfolding proteins and prevent the accumulation of insoluble aggregates because they have a higher proteasome activity than iSpMNs.

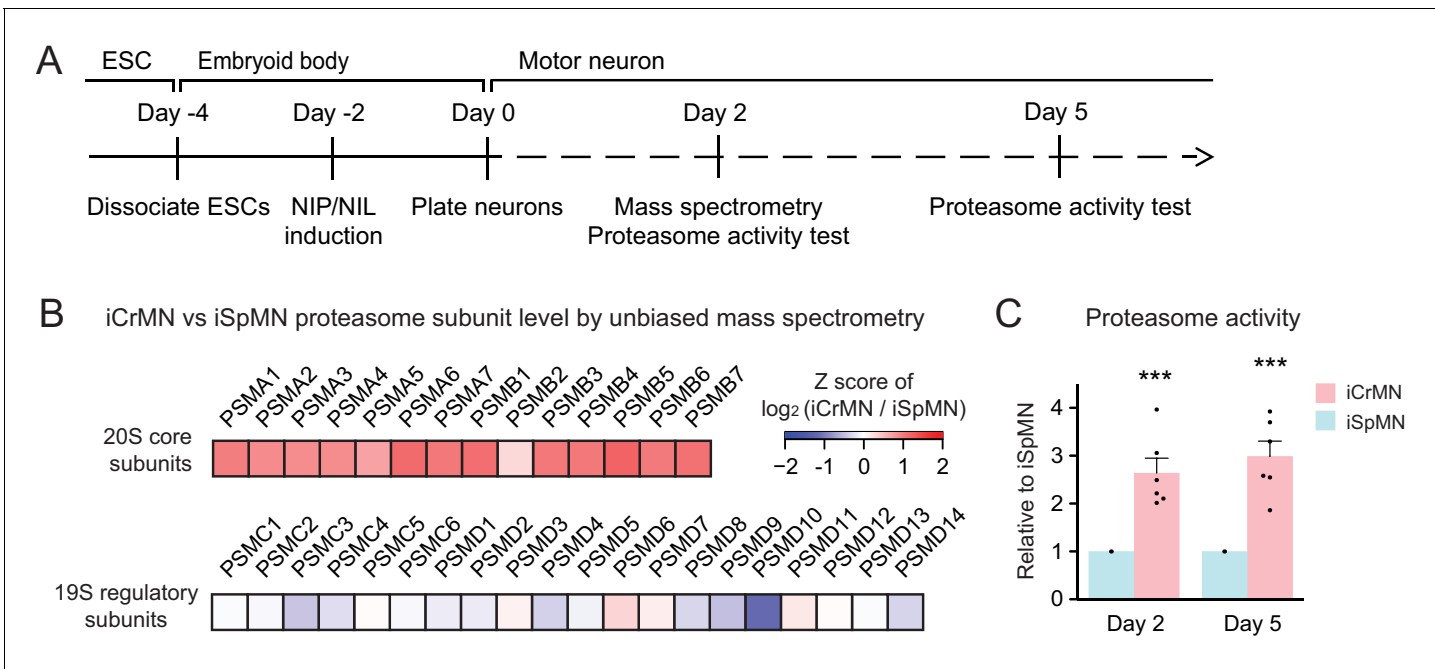

**Figure 9.** iCrMNs contain more proteasome 20S core subunits and have higher proteasome activity than iSpMNs. (**A**) Experimental outline: mass spectrometry was performed in Day two neurons and proteasome activity tests were performed in Day two and Day five living neurons. (**B**) Unbiased mass spectrometry revealed that iCrMNs contain higher levels of all proteasome 20S core subunits but not 19S regulatory subunits (n = 2). Proteomic data are available in *Figure 9—source data 1*. (**C**) iCrMNs have a higher proteasome activity than iSpMNs measured by hydrolysis rate of Suc-LLVY-AMC in living cells (n = 6, mean ± SEM). n = biological replicates; statistical analysis was performed in log-transformed data by student's t-test, *p<0.05, **p<0.01, ***p<0.001.
DOI: https://doi.org/10.7554/eLife.44423.015

The following source data is available for figure 9:

**Source data 1.** Proteomic data of non-transgenic iCrMNs and iSpMNs by unbiased mass spectrometry.
DOI: https://doi.org/10.7554/eLife.44423.016

## Chemical and genetic proteasome activation rescues iSpMN sensitivity to proteostatic stress

The ultimate goal of comparing two types of neurons with a differential sensitivity to ALS was to first identify an intrinsic feature that varied between the neuronal subtypes and to then modify the feature so as to confer resistance upon ALS-sensitive neurons. iCrMNs are more resistant to proteostatic stress and seem to take better advantage of the proteasome pathway to degrade misfolding proteins. Thus, we hypothesized that enhancing proteasome activity would improve iSpMN's ability to cope with proteostatic stress. To test this hypothesis, we investigated if the p38 MAPK inhibitor PD169316, which had been shown to enhance proteasome activity in primary mouse neurons (*Leestemaker et al., 2017*), could modify iCrMN's and iSpMN's ability to survive CPA-induced proteostatic stress.

First, we confirmed that PD169316 enhances proteasome activity in ESC-derived motor neurons (*Figure 10A*). In agreement to what has been described in other neuronal cell types, PD169316 treatment increased chymotrypsin-like activity of the 20S proteasome in iSpMNs and iCrMNs in a dosage dependent manner (*Figure 10B*). Second, because inhibiting p38 can cause differential survival phenotypes in these motor neurons, we quantified its effect on iSpMNs and iCrMNs. 0.25–2 µM PD169316 did not affect the survival of iCrMNs or iSpMNs in culture (*Figure 10C*). Finally, we tested if PD169316 could rescue iSpMN sensitivity to CPA at concentrations that do not affect survival of motor neurons under normal conditions. Therefore, Day 2 ESC-derived motor neurons were co-treated with 10 µM CPA and 0.25–2 µM PD169316 for three days and their survival was quantified on Day 5 (*Figure 10A*). In a concentration dependent manner, co-treatment of PD169316 improved the survival of iSpMNs from $66 \pm 4\%$ with no PD169316% to $80 \pm 4\%$ with 2 µM PD169316 (*Figure 10D*). 0–2 µM PD169316 treatment did not significantly enhance iCrMN survival to 10 µM CPA (*Figure 10D*). Notably, enhancing proteasome activity by 2 µM PD169316 was sufficient to increase iSpMN survival to phenocopy the basal iCrMN resistance to 10 µM CPA treatment (indicated by the red dash line in *Figure 10D*).

To avoid possible off target effects of a chemical treatment and to confirm that the survival rescue by PD169316 was caused by direct regulation of the proteasome, we next took a genetic approach. PSMB5 encodes the 20S core catalytic subunit β5 in the proteasome, which displays chymotrypsin-like activity and is expressed at a higher level in iCrMNs than in iSpMNs (*Figure 9B*). PSMB5 overexpression had been shown to upregulate and increase the activity of all three catalytic β subunits and lead to integration of free α subunits into de novo assembled proteasomes, increasing proteasome activity and fibroblast survival to oxidative stress (*Chondrogianni et al., 2005*). To activate the proteasome by PSMB5 expression, post-mitotic Day 1 iSpMNs and iCrMNs were infected with lentiviruses expressing either LacZ control or PSMB5. On day 4, neurons were treated with DMSO or 30 µM CPA. Survival of control or PSMB5-expressing motor neurons was measured at Day 7 (*Figure 10E*). These experiments revealed that PSMB5 expression significantly improved iSpMN survival to CPA treatment compared to LacZ infected controls but showed no effects in iCrMNs (*Figure 10F*).

Together, these results demonstrate that iCrMNs are intrinsically more resistant than iSpMNs to proteostatic stress caused by misfolded proteins and this sensitivity can be partially rescued by chemical and genetic activation of the proteasome (*Figure 11*).

## Discussion

Differential vulnerability is a landmark of neurodegenerative diseases. However, access to large numbers of neuronal subtypes for biochemical and genetic analysis is restricted to a few neuronal classes. Differentiation strategies from stem cells, or even terminally differentiated cell types, promises to bridge this gap by producing neuronal fates that recapitulate key features of embryonically derived neurons. We took advantage of a rapid and efficient ESC differentiation platform that relies on transcription factor expression to generate two types of motor neurons, which not only express key fate markers but also model differential sensitivity to proteostatic stress. Thus, the established ESC-derived motor neuron differentiation and culture conditions can form the basis of an in vitro analysis platform to rapidly screen for genes and chemicals that affect the ability of motor neurons to maintain a healthy proteome.

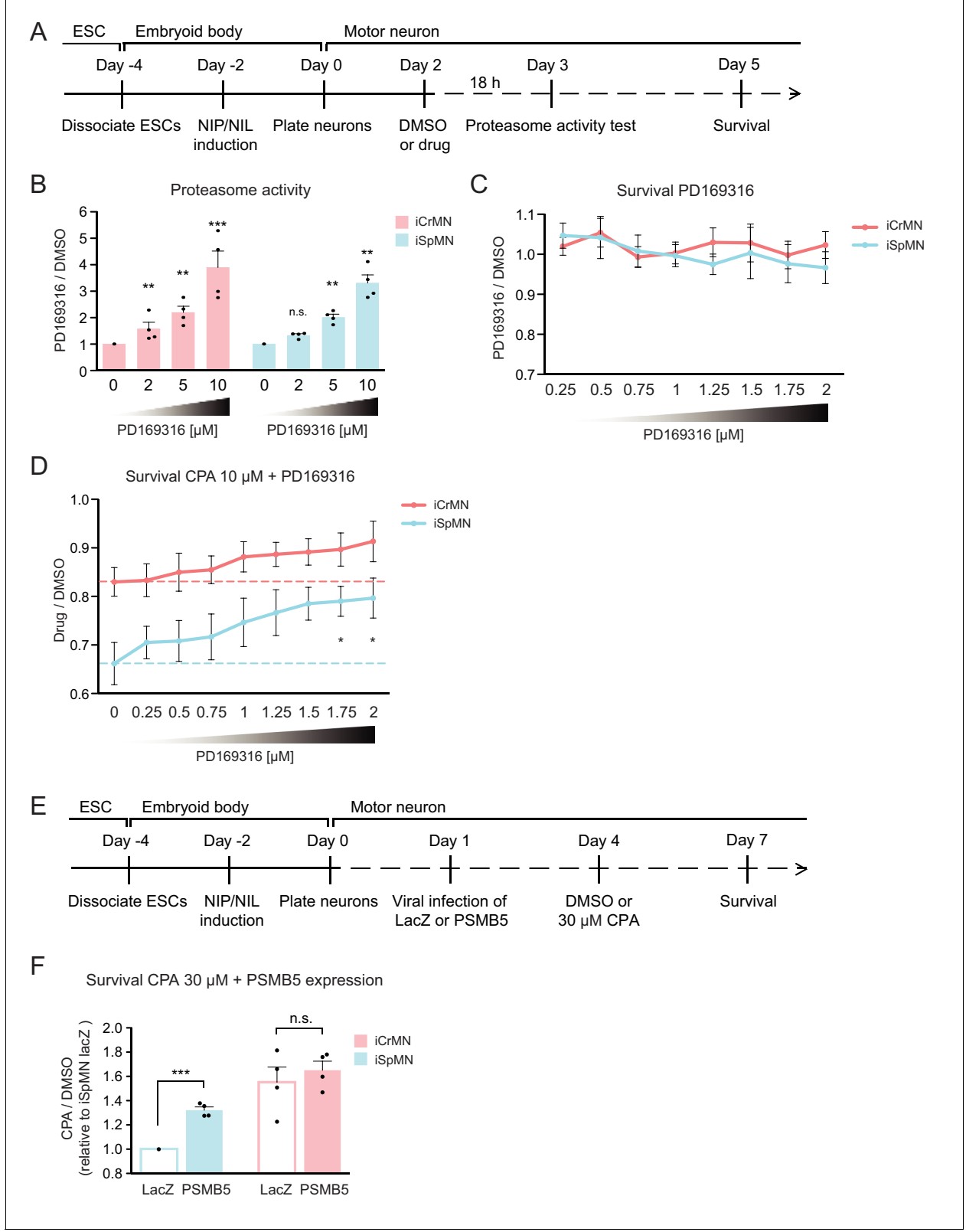

**Figure 10.** Chemical and genetic activation of the proteasome significantly increased iSpMN survival to proteostatic stress. (**A**) Experimental outline: PD169316 effect on proteasome was measured by proteasome activity test after 18 hr treatment of 0–10 μM PD169316 in Day two neurons. Neurons for survival assay were co-treated with 10 μM CPA and DMSO or PD169316 from Day two to Day five and the number of living cells were measured on Day 5. (**B**) PD169316 increases proteasome activity in both iCrMNs and iSpMNs (n = 4, mean ± SEM). (**C**) Treatment of 0.25–2 μM PD169316 had no effect on

*Figure 10 continued on next page*

*Figure 10 continued*

iCrMN and iSpMN survival (n = 6, mean ± SEM). (**D**) Co-treatment of 1.75 or 2 µM PD169316 with 10 µM CPA significantly increased iSpMN survival compared to treatment of 10 µM CPA alone. There is no significant difference between iCrMN survival with 10 µM CPA alone and iSpMN survival with 10 µM CPA and 0.75–2 µM PD169316 (n = 6, mean ± SEM). (**E**) Experimental outline: Day 1 iSpMNs and iCrMNs were infected with lentiviruses expressing either LacZ control or PSMB5 and then treated with DMSO or 30 µM CPA on Day four and assessed survival three days later on Day 7. (**F**) Viral expression of PSMB5 significantly increased iSpMN survival to CPA treatment (n = 4, mean ± SEM). The CPA versus DMSO survival ratio of iSpMNs with LacZ expression was used for normalization for all samples. n = biological replicates; statistical analysis was performed by student's t-test, *p<0.05, **p<0.01, ***p<0.001.

DOI: https://doi.org/10.7554/eLife.44423.017

During embryonic and in vitro differentiation, SpMNs and CrMNs are established by unique transcription factor combinations that induce different molecular and cellular properties in differentiated neurons. Our data suggest that, among these differences, the differential use of proteasome-dependent protein degradation is critical for the variable accumulation of insoluble aggregates between iCrMNs and iSpMNs. iCrMNs rely more than iSpMNs on proteasome function to degrade misfolded

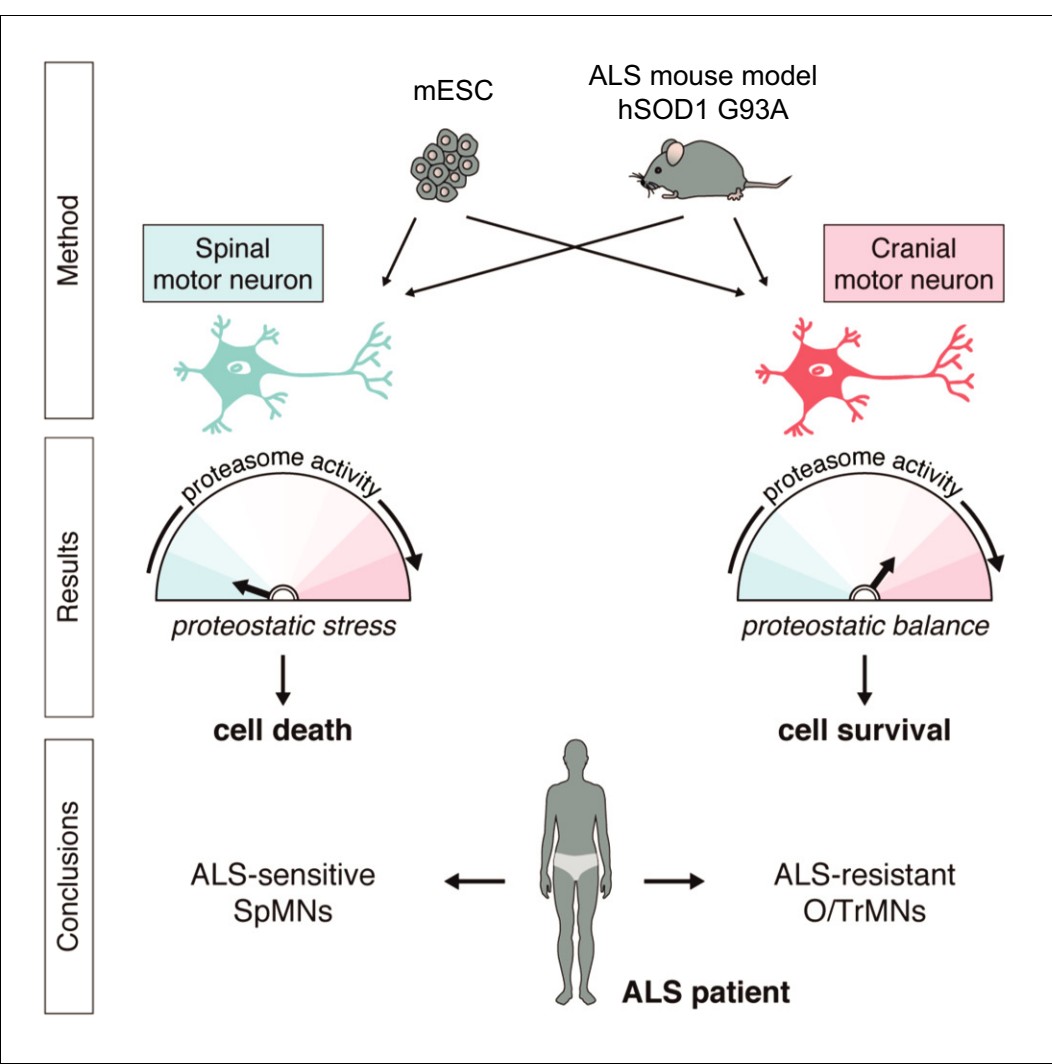

**Figure 11.** Summary. Differential vulnerability study using this platform equipped with in vitro and in vivo derived CrMNs and SpMNs has identified a superior proteostatic capacity to maintain a healthy proteome as a possible mechanism to resist ALS-induced neurodegeneration.

DOI: https://doi.org/10.7554/eLife.44423.018

proteins and minimize aggregate formation and accumulation. Importantly, this superior capacity of iCrMNs to handle misfolded proteins is not induced by hSOD1 mutant expression and is not specific to misfolded hSOD1. This capability is likely to be an intrinsic mechanism of iCrMNs that applies to general misfolded proteins ubiquitinated for proteasome degradation. This notion is supported by the differential survival of non-transgenic iCrMNs and iSpMNs under proteostatic stress caused by tunicamycin and CPA induced misfolded proteins. The proteomic study and the proteasome activity test in non-transgenic iCrMNs and iSpMNs suggest iCrMNs' higher proteasome activity is one of the mechanisms that make them resistant to proteostatic stress. A recent study utilizing a slightly different strategy but also forcing Phox2a expression reported that stem cell-derived CrMNs were more resistant to kainic acid toxicity than SpMNs (*Allodi et al., 2019*). ALS resistant motor neurons might utilize more than one mechanism to resist ALS, and future studies will be necessary to elucidate if they act in parallel during ALS progression. On the other hand, different CrMN features could be exploited to resist different ALS-causing mutations or sporadic ALS.

Activation of the proteasome by the p38 MAPK inhibitor PD169316 improved iSpMN survival to CPA induced proteostatic stress. In hSOD1 G93A mice, persistent activation of p38 was observed in motor neurons beginning in the presymptomatic stages of the disease (*Tortarolo et al., 2003*; *Holasek et al., 2005*; *Morfini et al., 2013*). Chemical inhibition of p38 enhanced SpMN survival and rescued axonal transport defects (*Dewil et al., 2007*; *Gibbs et al., 2018*). Our data now suggest that these protective effects, via the treatment of p38 inhibitors, may also benefit from enhanced activity of the proteasome and thereby improved proteostasis of misfolded proteins. Thus, a screen to identify more potent activators of the proteasome degradation pathway or to augment the UPR response might be a promising future therapeutic approach to ALS (*Das et al., 2015*). p62 also serves as a selective autophagy receptor for protein aggregates (*Dikic, 2017*). The less insoluble p62 accumulation in iSpMNs compared to iCrMNs after proteasome perturbation implies that iSpMNs may rely more on autophagy to clear p62-containing aggregates than iCrMNs. However, we found that treatment of autophagy inhibitors 3-Methyladenine and Bafilomycin A1 did not significantly alter aggregate accumulation in Day 5 iSpMNs and iCrMNs expressing hSOD1 mutant. Considering the complicated cross-talk between the autophagy and proteasome pathways, we also tested co-treatment of these autophagy inhibitors together with MG-132 but still observed no significant changes compared to MG-132 treatment alone. However, we are not confident to conclude that autophagy does not play a significant role protecting CrMNs simply relying on negative results using chemical inhibitors of the autophagy pathways.

One challenge of studying neurodegenerative diseases in vitro and for this ESC-based platform in particular is trying to model later aspects of ALS. Late and end-stage features of the disease can take several decades to develop in patients and months to develop in mice. In this study, similar to ALS mouse models, we accelerated the phenotype by expressing high levels of hSOD1. The caveat of this strategy is revealed in the accumulation of insoluble p62 in neurons expressing WT hSOD1, which might be explained by the proteostatic stress induced by high hSOD1 protein levels (*Bosco et al., 2010*). In the future, this ESC-based platform could be enhanced by co-culturing motor neurons with astrocytes and other cell types that may play a role in ALS progression (*Yamanaka et al., 2008*; *Phatnani et al., 2013*; *Meyer et al., 2014*).

The pathogenic mechanisms behind ALS remain obscure. This, in part, is due to the considerable percentage of sporadic cases and the wide array of ALS-causing genes involved in distinct biological processes. Therefore, studying a single ALS-causing mutation in a single cell type might fall short in revealing a complete picture of the ALS-induced degeneration process. The combination of a rapid ESC-based approach and in vivo validation helps to identify the intrinsic differences that play a role in CrMN's resistance to proteostatic stress compared to SpMNs. These unique intrinsic characteristics of CrMNs and SpMNs are likely to contribute to their differential vulnerability to ALS neurodegeneration. Using neuronal types with varying ALS sensitivities but comparable genetic makeups helps to juxtapose the differences in their innate physiology and stress response. These distinct differences between CrMNs and SpMNs can then be used to help identify ALS-specific pathogenic mechanisms and develop therapeutic strategies with broader applications within neurodegenerative fields.

# Materials and methods

## Key resources table

| Reagent type (species) or resource | Designation | Source or reference | Identifiers | Additional information |
|---|---|---|---|---|
| Strain, strain background (Mus musculus) | 129S1/C57BL/6J IslMN:GFP | PMID: 18031680 | MGI: J:132726 | |
| Strain, strain background (Mus musculus) | C57BL/6J | The Jackson Laboratory | JAX: 000664 | |
| Strain, strain background (Mus musculus) | B6.Cg-Tg(SOD1* G93A)1Gur/J | The Jackson Laboratory | JAX: 004435 | |
| Cell line (Mus musculus) | iNIL ESC line | This paper; PMID: 23872598 | | Cell line maintained in Mazzoni lab |
| Cell line (Mus musculus) | iNIP ESC line | This paper; PMID: 23872598 | | Cell line maintained in Mazzoni lab |
| Cell line (Mus musculus) | hSOD1 WT #8 iNIL ESC line | This paper | | Cell line maintained in Mazzoni lab |
| Cell line (Mus musculus) | hSOD1 WT #8 iNIP ESC line | This paper | | Cell line maintained in Mazzoni lab |
| Cell line (Mus musculus) | hSOD1 A4V #15 iNIL ESC line | This paper | | Cell line maintained in Mazzoni lab |
| Cell line (Mus musculus) | hSOD1 A4V #15 iNIP ESC line | This paper | | Cell line maintained in Mazzoni lab |
| Cell line (Mus musculus) | hSOD1 G93A #14 iNIL ESC line | This paper | | Cell line maintained in Mazzoni lab |
| Cell line (Mus musculus) | hSOD1 G93A #14 iNIP ESC line | This paper | | Cell line maintained in Mazzoni lab |
| Antibody | anti-Tubb3 (mouse monoclonal) | BioLegend | Cat. #801201; RRID:AB_2313773 | (1:500) |
| Antibody | anti-Islet1 (rabbit monoclonal) | Abcam | Cat. #ab109517; RRID:AB_10866454 | (1:200) |
| Antibody | anti-HA (rabbit polyclonal) | Abcam | Cat. #ab9110; RRID:AB_307019 | (1:1000 for ICC; 1:5000 for WB) |
| Antibody | anti-SQSTM1/p62 (mouse monoclonal) | Abcam | Cat. #ab56416; RRID:AB_945626 | (1:500 for ICC; 1:5000 for WB) |
| Antibody | anti-ChAT (goat polyclonal) | Millipore | Cat. #AB144P; RRID:AB_2079751 | (1:100) |
| Antibody | anti-SOD1 (rabbit polyclonal) | Abcam | Cat. #ab16831; RRID:AB_302535 | (1:1000) |
| Antibody | anti-ubiquitin (mouse monoclonal) | Cell Signaling Technology | Cat. #3936; RRID:AB_331292 | (1:5000) |
| Antibody | anti-β-actin (mouse monoclonal) | Santa Cruz | Cat. #sc-47778; RRID:AB_2714189 | (1:5000) |
| Peptide, recombinant protein | LIF | Fisher | Cat. #ESG1107 | |
| Peptide, recombinant protein | GDNF | PeproTech | Cat. #450–10 | |

*Continued on next page*

*Continued*

| Reagent type (species) or resource | Designation | Source or reference | Identifiers | Additional information |
|---|---|---|---|---|
| Peptide, recombinant protein | BDNF | PeproTech | Cat. #450–02 | |
| Peptide, recombinant protein | CNTF | PeproTech | Cat. #450–13 | |
| Peptide, recombinant protein | Suc-LLVY-AMC | Enzo Lifesciences | Cat. #BML-P802 | |
| Commercial assay or kit | In-Fusion HD Cloning Kit | Clontech | Cat. #639647 | |
| Commercial assay or kit | Mouse ES Cell Nucleofector Kit | Lonza | Cat. #VVPH-1001 | |
| Commercial assay or kit | RNeasy Mini Kit | Qiagen | Cat. #74104 | |
| Chemical compound, drug | CHIR99021 | BioVision | Cat. #1991 | |
| Chemical compound, drug | PD0325901 | Sigma | Cat. #PZ0162 | |
| Chemical compound, drug | Forskolin | Fisher | Cat. #BP2520-5 | |
| Chemical compound, drug | Isobutylmethylxanthine (IBMX) | Tocris | Cat. #2845 | |
| Chemical compound, drug | 5-Fluoro-2′-deoxyuridine | Sigma | Cat. #F0503 | |
| Chemical compound, drug | Uridine | Sigma | Cat. #U3003 | |
| Chemical compound, drug | Doxycycline | Sigma | Cat. # D9891 | |
| Chemical compound, drug | Cyclopiazonic acid (CPA) | Sigma | Cat. #C1530 | |
| Chemical compound, drug | Tunicamycin | Sigma | Cat. #T7765 | |
| Chemical compound, drug | Thapsigargin | Sigma | Cat. #586005 | |
| Chemical compound, drug | Brefeldin A | Sigma | Cat. #B7651 | |
| Chemical compound, drug | MG-132 | Sigma | Cat. #474790 | |
| Chemical compound, drug | PD169316 | Sigma | Cat. #P9248 | |
| Software, algorithm | Seurat version 3.0 | PMID: 25867923; 29608179 | | |
| Software, algorithm | Proteome Discoverer 2.2 | Thermo Fisher Scientific | | |
| Other | Fluorescein diacetate | Sigma | Cat. #F7378 | Used as cell viability stain |

## ESC lines

The inducible mouse ESC lines were generated based on the inducible cassette exchange (ICE) system (*Iacovino et al., 2011*). Transcription factor cassettes NIL (Ngn2-Isl1-Lhx3) and NIP (Ngn2-Isl1-Phox2a) were used to program spinal and cranial motor neurons respectively as previously described (*Mazzoni et al., 2013*) but with different cloning methods. Ngn2-Isl1-Lhx3 and Ngn2-Isl1-Phox2a coding sequences (linked by 2A peptide sequences) were cloned from previous plasmids (*Mazzoni et al., 2013*) and directly inserted into the p2lox-V5 plasmid using In-Fusion HD Cloning Kit by homologous recombination (Clontech, 639647). In the generated p2lox-NIL-V5 and p2lox-

NIP-V5 plasmids, Lhx3 and Phox2a in the NIL and NIP cassettes were tagged by V5-His double epitope. Phusion High-Fidelity polymerase (Thermo Fisher, F531S) was used during PCR amplification.

Isogenic inducible lines iNIL and iNIP were generated from A17 iCre ESCs from PMCID: PMC3820498 (*Mazzoni et al., 2013*). The ESCs are routinely tested for mycoplasma. The same batch of A17 iCre ESCs were treated for 14 hr with 1 µg/ml doxycycline to induce Cre expression followed by nucleofection (Mouse ES Cell Nucleofector Kit, Lonza VVPH-1001) of the respective plasmids. 400 µg/ml G418 (Sigma A1720-1G) was added after 24 hr for selection. After one week of selection, cell lines were characterized by performing immunostaining of V5 antibody (mouse monoclonal, Thermo Fisher Scientific R960-25) and expanded.

To obtain isogenic iNIL and iNIP lines with same hSOD1 transgenes, A17 iCre human SOD1 (hSOD1) transgenic lines were first produced using the Tol2 transposon system. Addgene plasmid 59740 pTol2-CAGGS::ChR2-YFP was used as the backbone, and ChR2-YFP was replaced by the coding sequences of hSOD1WT, hSOD1A4V and hSOD1G93A cloned from addgene plasmids 26397, 26398 and 26401 respectively. HA epitope tag was added to the 3' end of hSOD1 coding sequences during PCR amplification. The generated pTol2-CAGGS::hSOD1-HA plasmids (pTol2-CAGGS:: hSOD1WT-HA, pTol2-CAG::hSOD1A4V-HA and pTol2-CAG::hSOD1G93A-HA) contained a pCAGGS promoter, a hSOD1-HA transgene and a Zeo selection marker, which were all flanked by two Tol2 sequences. To make the A17 iCre hSOD1 transgenic lines, Tol2 transposase transient expression plasmid pCMV-Tol2 (addgene 31823) and respective pTol2-CAGGS::hSOD1-HA plasmid were co-nucleofected into the Dox primed cells as described above. After one week of selection with 50 µg/mL Zeocin (Invivogen ant-zn-1), 10–20 clones were picked, analyzed by immunostaining using HA antibody (rabbit polyclonal, Abcam ab9110) and expanded for each line. hSOD1-HA transgene expression level in these clones were quantified by Western Blot analysis using HA and SOD1 antibodies (rabbit polyclonal, Abcam ab16831) and the clones expressing the desired amount of hSOD1 proteins were used later to generate isogenic iNIL and iNIP lines.

All mouse ESC lines were cultured in 2-inhibitor based medium (Advanced DMEM/F12:Neurobasal (1:1) medium (Thermo Fisher, 12634028, 10888022) supplemented with 2.5% ESC-grade Tet-negative fetal bovine serum (vol/vol, VWR 35–075-CV), 1X N2 (Thermo Fisher, 17502–048), 1X B27 (Thermo Fisher, 17504–044), 2 mM L-glutamine (Thermo Fisher, 25030081), 0.1 mM β-mercaptoethanol (Thermo Fisher, 21985–023), 1000 U/ml leukemia inhibitory factor (Fisher, ESG1107), 3 µM CHIR99021 (Biovision, 1991) and 1 µM PD0325901 (Sigma, PZ0162-5MG), and maintained at 37°C, 8% $CO_2$.

## ESC-derived motor neurons

To differentiate ESCs into motor neurons, ESCs were trypsinized (Thermo Fisher, 25300–120) and seeded as single cells at 25,000 cells/ml in ANDFK medium (Advanced DMEM/F12: Neurobasal (1:1) Medium (Thermo Fisher, 12634028, 10888022), 10% Knockout SR (vol/vol) (Thermo Fisher, 10828–028), 2 mM l-glutamine (Thermo Fisher, 25030081), and 0.1 mM 2-mercaptoethanol (Thermo Fisher, 21985–023)) to initiate formation of embryoid bodies (EBs) (Day −4) in the suspension culture using 10 cm untreated dishes (Fisher, 08-772-32) and maintained at 37°C, 5% $CO_2$. Medium was changed 2 days later (Day −2) with addition of 3 µg/mL Doxycycline (Sigma D9891) to induce the transcription factors NIL and NIP. The EBs treated with Doxycycline for 2 days were dissociated by 0.05% Trypsin-EDTA (Thermo Fisher, 25300–120) on Day 0. The dissociated neurons were plated with a density of 60,000/mL in 24-well plates (Fisher, 142475) for survival assay or 1 million/mL in a 10 cm dish (Fisher, 12-567-650) for Western Blot. The plates are coated by 0.001% poly-D-lysine (Sigma P0899-10MG) overnight and washed with sterile water twice.

All ESC-derived motor neurons were grown in motor neuron medium (Neurobasal medium (Thermo Fisher, 10888022) supplemented with 2% fetal bovine serum (vol/vol, VWR 35–075-CV), 1X B27 (Thermo Fisher, 17504–044), 0.5 mM L-glutamine (Thermo Fisher, 25030081), 0.01 mM β-mercaptoethanol (Thermo Fisher, 21985–023), 10 ng/mL GDNF (PeproTech 450–10), 10 ng/mL BDNF (PeproTech 450–02), 10 ng/mL CNTF (PeproTech 450–13), 10 µM Forskolin (Fisher BP2520-5) and 100 µM Isobutylmethylxanthine (IBMX) (Tocris 2845)), and maintained at 37°C, 5% $CO_2$. 3 µg/mL doxycycline (Sigma D9891) was used to maintain transcription factor expression. 4 µM 5-Fluoro-2'-deoxyuridine (Sigma F0503) and 4 µM uridine (Sigma U3003) were used to kill proliferating cells that failed in motor neuron programming.

## Animals

129S1/C57BL/6J *Isl^MN:GFP* mice (MGI: J:132726) were used for visualization and isolation of *Isl^MN: GFP* labeled embryonic motor neurons. p62 and SOD1 staining in lumbar spinal cord and oculomotor neurons was performed on tissue from first generation crosses of C57BL/6J females and *SOD1*^G93A transgenic male mice [C57BL/6J, 000664; B6.Cg-Tg(SOD1*G93A)1Gur/J, 004435, Jackson Laboratory]. All experiments utilizing laboratory animals were performed in accordance with NIH guidelines for the care and use of laboratory animals, and with approval of the Institutional Animal Care and Use Committees of Boston Children's Hospital and Columbia University.

## Primary motor neurons

To dissect primary motor neurons, E11.5 *Isl^MN:GFP*-positive embryos were harvested in ice-cold phosphate buffered saline (PBS; Thermo Fisher Scientific, 10010–023), and individually stored on ice in 24-well plates (Olympus Plastics brand manufactured by Genesee Scientific, San Diego, CA, USA, 25–107) filled with 2 ml Hibernate-E low fluorescence (BrainBits, LLC., Springfield, IL, USA, HELF) supplemented with 1X B27 (Thermo Fisher Scientific, 17504–044). Four regions were microdissected for each experiment: oculomotor/trochlear neurons (O/TrMNs); spinal motor neurons (SpMNs) from the cervical-lumbar spinal cord; GFP-positive control facial motor neurons; GFP-negative control extremities. Embryos were microdissected in ice-cold Hanks' Balanced Salt Solution (HBSS; Thermo Fisher Scientific, 14175–095) using fine forceps (Roboz Surgical Instrument Co, Inc, Gaithersburg, MD, USA, RS-5015) and a micro knife (Roboz, RS-6272) under a fluorescence dissecting microscope (Nikon, Tokyo, Japan, SMZ18 and SMZ1500). Dissected samples were then individually collected in 1.7 ml microcentrifuge tubes containing 500 µl of Hibernate-E (BrainBits, HE) supplemented with 1X B27, 2% Horse serum (Thermo Fisher Scientific, 26050–070), 1X GlutaMAX (Thermo Fisher Scientific, 35050–061), and 100 U/ml penicillin-streptomycin (Thermo Fisher Scientific, 15140–122) and stored on ice until dissociation step. Total dissection time per experiment was typically 4–5 hr.

To dissociate primary motor neurons, microdissected tissue samples were incubated in a papain solution (20 units/ml papain and 0.005% DNase; Papain Dissociation System, Worthington Biochemical Corp., Lakewood, NJ, USA, LK003150), with gentle agitation for 30 min at 37°C. Dissociated cells were then transferred and triturated in a DNAse/albumin-ovomucoid inhibitor solution to inactivate the papain. The cells were pelleted in a centrifuge at 300 g for 5 min, resuspended in Hibernate-E (HE) supplemented with 1X B27, 2% Horse serum, 1X GlutaMAX, and 100 U/ml penicillin-streptomycin, and stored on ice until cell sorting. Total dissociation time per experiment was typically 1.5 hr.

To purify primary motor neurons by fluorescent-activated cell sorting (FACS), single cell suspensions of each sample were prepared at approximately $1-9 \times 10^6$ cells/ml. GFP-positive motor neurons were isolated with a Becton Dickinson ARIA SORP or ARIA II SORP cell sorter. Data were analyzed with FacsDIVA software. FITC fluorescence was excited with a 488 nm laser and detected with a 530/30 nm (ARIA) or 525/50 nm (ARIA II) filter. To minimize cell damage, FACS was conducted using a ceramic 100 µm nozzle (Becton Dickinson), sheath pressure of 20 pounds per square inch (psi) and a low acquisition rate of 1,000–4,000 events/s. O/TrMN and SpMN cell samples were collected in 1.7 ml microcentrifuge tubes containing 500 µl of the culture medium (see below) and stored on ice until plating. Total FACS time per experiment was typically 1–2 hr.

96-well plates (Greiner Bio-One International, Kremsmünster, Austria, 655090) were coated with 20 µg/ml poly D-lysine (EMD Millipore, Burlington, MA, USA, A-003-E) in PBS overnight at 37°C. Wells were washed ×3 with sterilized water, then coated with 10 µg/ml laminin (Thermo Fisher Scientific, 23017–015) in PBS overnight at 37°C. FACS-isolated primary O/TrMNs and SpMNs were plated at densities of 1000 and 2,000 cells per well, respectively, in coated 96-well plates and cultured in neurobasal medium (Thermo Fisher Scientific, 12348–017) supplemented with 1X B27, 2% Horse serum, 1X GlutaMAX, 100 U/ml penicillin-streptomycin, 50 µM 2-mercaptoethanol (Sigma-Aldrich, M6250), 10 µM Forskolin (Thermo Fisher Scientific, BP25205), 100 µM IBMX (Tocris Cookson, Bristol, United Kingdom, 2845), and 10 ng/ml of BDNF, CNTF, and GDNF (ProSpec-Tany TechnoGene, Ltd., Rehovot, Israel, CYT-207, CY-272, and CYT-305, respectively). Half of the medium was refreshed every 5 days.

## Immunocytochemistry and imaging

For all media changes and staining steps, 100 µl of original medium was left to avoid detaching cultured cells. To compensate for this additional volume, we added an equal volume of double concentrated paraformaldehyde, permeabilizing solution, blocking solution, and primary and secondary antibody solution. Cells were fixed with 37°C 4% paraformaldehyde (Electron Microscopy Sciences, Hatfield, PA, USA, 15710)/sucrose (Sigma-Aldrich, S8501) in PBS for 30 min. Cells were washed with PBS ×3 and incubated for 5 min in permeabilizing solution [PBS containing 0.2% Triton X (Sigma-Aldrich)]. After additional PBS washes ×3, cells were incubated for 1 hr in blocking solution [PBS containing 5% bovine serum albumin (Sigma-Aldrich, A4503) and 0.05% Tween (Sigma-Aldrich, P1379)]. Subsequently, cells were incubated with primary antibodies for 1 hr at room temperature or overnight at 4°C, washed in PBS ×3, incubated with fluorescence-conjugated secondary antibodies for 1 hr at room temperature or overnight at 4°C, washed in PBS ×3, and nuclei were counterstained with 4′,6-diamidino-2-phenylinodole (DAPI; Thermo Fisher Scientific, D1306). Cells were then rinsed in PBS ×2 and in filtered water ×3. Cell counting was performed and representative images were captured with a Nikon Perfect Focus Eclipse Ti live cell fluorescence microscope using Elements software (Nikon) and 10X (for acquiring images) and 20X (for counting cells) objectives. Samples were imaged and processed to achieve maximum signal intensity without saturated pixels.

## Antibodies and reagents

Reagents used in the western blot and survival assays: cyclopiazonic acid (CPA; Sigma, C1530-5MG), tunicamycin (Sigma, T7765), thapsigargin (Sigma, 586005), brefeldin A (Sigma, B7651), MG-132 (Sigma, 474790), PD169316 (Sigma, P9248) and dimethyl sulfoxide (DMSO; Sigma, D2650).

### Primary antibodies used for immunocytochemistry (ICC)

mouse monoclonal antibody to neuronal class III β-tubulin (Tuj1, 1:500; BioLegend, MMS-); rabbit monoclonal antibody to Islet1 (1:200; Abcam, ab109517); rabbit polyclonal antibody to HA tag (1:1000; Abcam, ab9110); mouse monoclonal antibody to SQSTM1/p62 (1:500; Abcam, ab56416) and goat polyclonal anti-ChAT (1:100; AB144P, Millipore).

### Secondary fluorescence-conjugated antibodies for ICC

Alexa Fluor 488-, 562- or 594-conjugated goat anti-mouse IgG (1:400 or 1:1000; Thermo Fisher Scientific, Inc, Waltham, MA, USA, A-11001); 488- or 594-conjugated goat anti-rabbit IgG (1:400 or 1:1000; Thermo Fisher Scientific, A-11072); for mouse tissue section: donkey anti-mouse and anti-goat secondary antibodies (1.5 µg/mL; Jackson ImmunoResearch).

### Primary antibodies used for western blot

rabbit polyclonal antibody to HA tag (1:5000; Abcam, ab9110); mouse monoclonal antibody to SQSTM1/p62 (1:5000; Abcam, ab56416); rabbit polyclonal antibody to Superoxide Dismutase 1 (SOD1) (1:1000; Abcam, ab16831); mouse monoclonal antibody to ubiquitin (1:5000; CST, #3936); mouse monoclonal antibody to beta-actin (1:5000; Santa Cruz, sc-47778).

### Secondary antibodies used for Western blot

HRP-conjugated mouse and rabbit secondary antibodies (Amersham ECL Western Blotting Detection Reagent, VWR, 95038–560).

## Survival assay of ESC-derived motor neurons

Neurons were washed with 1X HBSS (Thermo Fisher, 14025092) and viable cells were stained by 0.1 µg/mL fluorescein diacetate (Sigma, F7378) and tile-imaged (~0.3 cm$^2$) by Nikon fluorescent microscope at 10X. The number of living cells was counted by the spot detection function of Nikon software. Each biological replicate had at least three technical replicates and the average was used for analysis.

## Western blot assays

Cells were lysed in RIPA buffer (Sigma, R0278) with protease inhibitors (Sigma, 11697498001) for 30 min on ice. After centrifuge at 14,000 g for 15 min at 4°C, the supernatant was collected as soluble fraction and the pellet was solubilized using 8M urea with 2% SDS and 50 mM pH7.6 Tris-HCl buffer as insoluble fraction. 5–15 µg protein sample from soluble fractions and equivalent volumes of insoluble fractions were separated by 8–16% SDS-PAGE (Bio-Rad, 456–1105) and transferred to PVDF membranes (Thermo Fisher, 88518). The membrane was blocked with 5% Bovine Serum Albumin (Sigma, A9647) for 1 hr at room temperature, incubated with primary antibodies overnight at 4°C, washed in TBST X3, incubated with HRP-conjugated mouse and rabbit secondary antibodies (Amersham ECL Western Blotting Detection Reagent, VWR, 95038–560) for 1 hr at room temperature, washed in TBST X3 and then developed using ECL Western Blotting Detection System (Fisher Scientific, RPN2108) and X-ray films (Thermo Fisher, 34091).

## RNA preparation and RT-qPCR

RNA was extracted using QIAGEN RNeasy kit following manufacturer's instructions. For quantitative PCR analysis, cDNA was synthesized using SuperScript III (Invitrogen), amplified using Maxima SYBR green brilliant PCR amplification kit (Thermo Scientific) and quantified using a CFX 96 Touch Biorad qPCR thermocycler (Biorad). One independent differentiation was considered to be a biological replicate (n = 1).

## Proteasome activity test

For quantifying MG-132 effect on proteasome activity, cell lysates were collected from Day 4 neurons with 6 hr treatment of 100 nM MG-132. The measurement was carried out in a total volume of 100 µL in 96 well plates with 50 µM Suc-LLVY-AMC (BML-P802, Enzo Lifesciences) in assay buffer (25 mM HEPES, 0.5 mM EDTA, 0.05% NP-40 Tergitol and 0.01% SDS) using 5, 15 and 25 µg of protein supernatant. The chymotrypsin-like activity of proteasome was determined by increase in AMC fluorescence per minute per µg of protein supernatant and compared between MG-132 and DMSO treated samples. Fluorescence was measured every 10 min for 120 min at 37°C using a Tecan M200 Microplate Reader (Tecan; λex/em = 380/460 nm).

For measuring proteasome activity in living cells with DMSO or PD169316 treatment, cells were plated in 96 well plates, washed and added assay buffer (20 mM Tris, 5 mM MgCl2, 1 mM DTT, 1 mM ATP, pH = 7.4) containing 100 µM Suc-LLVY-AMC in a total volume of 150 µL. Cell density was subsequently measured for normalization. The chymotrypsin-like activity of proteasome was determined by increase in AMC fluorescence per minute per cell and compared between PD169316 and DMSO treated samples or between iCrMNs and iSpMNs. Fluorescence was measured every 5 min for 50 min at 37°C using a Tecan Spark Microplate Reader (Tecan; λex/em = 380/460 nm).

## Proteomics analysis by mass spectrometry

We collected two different types (iSpMNs and iCrMNs) of neuronal cells form two different time points (0 hr and 12 hr) at day2. For proteomics analysis, we processed cells from two different biological replicates. We resuspended cell pellets in 200 µL ice-cold 0.1% Rapigest in 50 mM TEAB containing 1:100 protease inhibitor cocktail. The cells were then sonicated with probe sonicator for 2 × 30 s with amplitude 5. The samples were returned to ice for 20 s between sonication intervals. It was then reduced in 15 mM DTT at 55°C for 45 min, and alkylated in 55 mM iodoacetamide in the dark at room temperature for 30 min. Finally, we used 1:20 (w/w) µg of mass spectrometry grade trypsin (Sigma Aldrich) to digest the proteins into peptides at 37°C overnight. We performed Hypersep (Thermo Fisher Scientific) cleanup and the purified peptides were vacuum dried using Vacufuge Plus (Eppendorf).

We measured peptide concentrations with the Pierce Quantitative Fluorometric Peptide Assay (Thermo Fisher) kit before labeling with Tandem mass tag (TMT) reagents (Thermo Scientific). We used four different TMT tags (126,127N, 129N, 129C) for two different time points of each cell type. The tags were dissolved in anhydrous acetonitrile (0.8 mg/40 µL) according to manufacturer's instruction. We labeled 30 µg peptide per sample labeled with 41 µL of the TMT reagent at final acetonitrile concentration of 30% (v/v). Following incubation at room temperature for 1 hr, we quenched the reactions with 8 µL of 5% hydroxylamine for 15 min. All samples were combined in a

new microcentrifuge tubes at equal amounts and vacuum dried to remove acetonitrile in vacuum concentrator (Eppendorf).

TMT-labeled tryptic peptides were subjected to high-pH reversed-phase high performance liquid chromatography fractionation using an Agilent 1200 Infinity Series with a phenomenex Kinetex 5 u EVO C18 100A column (100 mm x 2.1 mm, 5 mm particle size). Mobile phase A was 20 mM ammonium formate, and B was 90% acetonitrile and 10% 20 mM ammonium formate. Both buffers were adjusted to pH 10. Peptides were resolved using a linear 120 min 0–40% acetonitrile gradient at a 100 uL/min flow rate. Eluting peptides were collected in 2 min fractions. We combined about 72 fractions covering the peptide-rich region to obtain 24 samples for analysis. To preserve orthogonality, we combined fractions across the gradient that is each of the concatenated samples comprising fractions which were 12 fractions apart. Re-combined fractions were reduced using vacuum concentrator (Eppendorf), desalted with C18 stage-tip, and suspended in 95% mass spectrometry grade water, 5% acetonitrile, and 0.1% formic acid for subsequent low pH chromatography and tandem mass spectrometry analysis.

We used an EASY-nLC 1200 coupled on-line to a Q Exactive HF spectrometer (both Thermo Fisher Scientific). Buffer A (0.1% FA in water) and buffer B (80% acetonitrile, 0.5% acetic acid) were used as mobile phases for gradient separation. An EASY Spray 50 cm x 75 µm ID PepMap C18 analytical HPLC column with 2 µm bead size was used for peptide separation. We used a 110 min linear gradient from 5% to 23% solvent B (80% acetonitrile, 0.5% acetic acid), followed by 20 min from 23% to 56% solvent B, and 10 min from 56% to 100% solvent B. Solvent B was held at 100% for another 10 min. Full MS scans were acquired with a resolution of 120,000, an AGC target of 3e6, with a maximum ion time of 100 ms, and scan range of 375 to 1500 m/z. Following each full MS scan, data-dependent (Top 15) high resolution HCD MS/MS spectra were acquired with a resolution of 60,000, AGC target of 2e5, maximum ion time of 100 ms, 1.2 m/z isolation window, fixed first mass of 100 m/z, and NCE of 32.

We used Proteome Discoverer 2.2 (Thermo Fisher Scientific) with its integrated search engine Sequest HT to analyze our raw files acquired from the mass spectrometer. Data were searched against the mouse sequence file (Mus_musculusgrcm38.p6) downloaded from the ENSEMBL database. All sample fractions of two individual sets were grouped by setting up study factor design section of proteome discoverer. We set up the quantification using the parameters given for each TMT isotopes in certificate of analysis. The mass tolerance of MS/MS spectra were set to 20 ppm with a posterior global FDR of 1% based on the reverse sequence of the mouse FASTA file. In addition, MS/MS data were searched by Andromeda for potential common mass spectrometry contaminants. Trypsin/P specificity was used to perform database searches, allowing two missed cleavages. Carbamidomethylation of cysteine residues and 10-plex TMT modifications on Lys and N-terminal amines were considered as a fixed modification, while oxidation of methionines and N-terminal acetylation were considered as variable modifications. TMT quantification was performed at MS2 level with default mass tolerance and other parameters. We then used the reporter ion intensities as estimates for protein abundance. A total of ~8700 protein groups were identified. Protein groups with no measurement among either replicate were then removed, as well as those identified by only one peptide in either replicate. After filtering, for each protein the geometric mean was calculated across all samples within one stress and the intensities were divided by this mean. The median of these ratios over all proteins was used as a size factor to account for differences in global sequence coverage between samples, similar to library size normalization for RNA sequencing and foot printing experiments. Surrogate variables were estimated and removed via linear modeling (SVA) to remove batch effects (*Leek and Storey, 2007*).

## Viral production and infection of motor neurons

Lentiviral vectors were constructed expressing either LacZ or PSMB5 under the control of a constitutive CMV promoter, followed by an IRES sequence and GFP to permit identification of positively infected cells. The viral vectors were packaged using 2nd generation packaging plasmids transfected into HEK-293T cells using polyethylenimine. Viruses were collected from HEK-293T media 3 days after transfection and filtered. ESC-derived motor neurons were then infected on Day one after dissociation with equal volumes of either LacZ or PSMB5 expressing viruses. Viral medium was changed after 16 hr and neurons were then treated with DMSO or 30 µM CPA on Day four and fixed and imaged on Day seven to measure survival. Survival was measured as a ratio of infected (GFP-positive)

over total (DAPI-positive) cells and plotted as the ratio of CPA over DMSO, normalized to control LacZ infected iSpMNs.

## Polysome profile analysis

Polysome profiling was performed as described previously by *Gandin et al. (2014)*. Briefly, 48h-doxycycline-treated embryonic bodies from both iCrMNs and iSpMNs were incubated in the presence of 100 µg/mL cycloheximide for 10 min at 37°C followed by centrifugation at 700 rpm for 3 min to pellet the embryonic bodies. We incubated the embryonic bodies further with cold PBS containing 100 µg/mL cycloheximide on ice for 5 min. Following removal of the PBS, 450 µL of hypotonic buffer (5 mM Tris-HCl [pH 7.5], 2.5 mM $MgCl_2$, 1.5 mM KCl, 1x protease inhibitor cocktail [EDTA-free], 100 µg/mL cycloheximide, 1 µM DTT, 100 units of RNase inhibitor and 0.5% Triton X-100) was added and transferred to a pre-chilled Eppendorf tube. After 5 min of incubation on ice, the lysate was centrifuged at 16,000 x g for 7 min at 4°C and the supernatant (~400 µL) was transferred to a new pre-chilled 1.5 mL tube. A total of 250 µg of RNA was overlaid on a 5–50% sucrose gradient (buffered in 50 mM Tris-HCl [pH 7.0], 10 mM $MgCl_2$, 25 mM KCl, 100 µg/mL cycloheximide) and sedimented by ultracentrifugation for 120 min at 222,228 x g (36,000 rpm) in a Beckman SW41Ti rotor at 4°C. During sucrose gradient analysis, absorbance was monitored at 254 nm while ~900 µL fractions were collected using a Brandel density gradient fractionation system. RNA from polysome fractions were pooled and extracted using RNA TRIzol method and purified through RNeasy mini kit (Qiagen, Cat. #74104).

## Single-cell RNA sequencing and data availability

ESC-derived iCrMNs and iSpMNs cultured for 2, 5 and 7 days were dissociated with 0.05% trypsin and returned to culture media with supplements plus 0.1 µg/mL fluorescein diacetate to stain living cells. Single viable cells were sorted using Fluorescence Activated Cell Sorting (Becton Dickinson FACSAria II) to single wells of 96-well fully-skirted Eppendorf PCR plates in 5 uL 1X Buffer TCL (QIA-GEN, 1070498) (without 2-Mercaptoethanol), covered, spun at 2600 rpm for 1 min, frozen on dry ice, and stored at −80°C. For each time point, the same plate was filled in the first four rows with iCrMNs and last four rows with iSpMNs. Primary O/TrMNs and SpMNs cultured for 7 days were individually patched and their cytoplasmic contents extracted and added into 96-well plates containing pre-frozen TCL buffer. In all plates, positions H1/H2 contained 50 control cells. While the iCrMNs/iSpMNs and primary O/TrMNs/SpMNs were collected at different institutions, the library preparations and sequencing were all performed according to the same protocol in the same locations.

Single cell library preparation of full-length transcripts (without unique molecular identifiers, UMI) was performed with a modified Smart-seq2 protocol (*Picelli et al., 2013*; *Trombetta et al., 2014*). A total of 562 single cells were analyzed.

Paired-end next generation sequencing of the multiplexed libraries was performed on an Illumina NextSeq500 platform. The iCrMNs and iSpMNs were multiplexed and sequenced together, and the primary O/TrMNs and SpMNs were sequenced in a separate batch, for an approximate depth of 1 million reads/cell. RSEM (v1.2) with STAR (v2.5.3a) was run to align reads to mouse mm10 (Ensemble GRCh38-1.2.0) and generate estimated expression level TPM matrices.

Single cell analysis was performed principally with Seurat version 3.0 (*Satija et al., 2015*; *Butler et al., 2018*). Original sample names were modified as follows: 'NIP_day2' = 'iCrMN_day2', 'NIP_day5' = 'iCrMN_day5', 'NIP_day7' = 'iCrMN_day7', 'OMN_culture' = 'O/TrMN', 'NIL_day2' = 'iSpMN_day2', 'NIL_day5' = 'iSpMN_day5', 'NIL_day7' = 'iSpMN_day7', 'SMN_culture' = 'SpMN'). Single-cell data were cleaned up as follows: control pooled samples H1/H2 from each plate were removed from the matrix, data were filtered with minimum and maximum gene expression thresholds of 5000 and 12,500 respectively, to remove poor-quality cells and doublets, and filtered with a maximum 'fractional mitochondrial genes' of 0.15 to remove damaged cells. After clean-up, 450 cells remained in the analysis (80% of input). Variable genes were detected with mean.cutoff (0.0125, 3) and dispersion.cutoff (0.5, Inf), resulting 3982 variable genes. The data were then log normalized and scaled to regress out unwanted sources of variation (nCount_RNA, percent.mito). Linear dimensional reduction (principal component analysis, PCA) was performed. The first 50 PCs were evaluated for significance via a combination of tools available in Seurat and PCs 1–10 and 14 were included in subsequent analyses. To visualize high-dimensional single cell data, we chose to implement tSNE

(perplexity of 18). Clustering was also performed iteratively from a resolution of 0.1 to 0.8. Batch effect and biological significance of individual principal components was evaluated based on component genes and dataset separation. These datasets were analyzed multiple times with different variables to identify persistent features of the cell populations.

The single cell RNA sequencing raw FASTQ files used in this study are deposited in the National Center for Biotechnology Information's Gene Expression Omnibus (GEO), accession GSE130938.

### Survival assay in primary motor neurons

Neurons were analyzed by immunofluorescence labeling with the neuronal marker Tuj1 and the motor neuron marker Islet1, and nuclei were counterstained with DAPI. Neuronal cell body death was assessed from pyknotic nuclear morphology and the presence of membrane swelling in the cell body. Neuronal processes that showed signs of beading and swelling were classified as degenerating processes. Cells with neither cell body death nor degenerating processes were counted as viable neurons (Tuj1$^+$) or motor neurons (Tuj1$^+$, Isl1$^+$). Survival ratios in the experiments adding endoplasmic reticulum (ER) stressors or the proteasome inhibitor MG132 were calculated as the number of healthy viable cells in the drug treated well divided by the number of viable cells in the vehicle (DMSO) only well. All viable cells in each well of the 96-well plate were manually counted under the microscope by one counter (JYL). Random samples were re-counted by a second counter (RF). Cell counts differed by an average of 5 cells/well (<1.6% of the total cells).

### Animal tissue collection, Immunofluorescence Staining and Imaging

$SOD1^{G93A}$ mutant mice and their non-transgenic littermates (first generation cross only) were deeply anesthetized using Avertin (tribromoethanol, Sigma) and fixed by transcardial perfusion with 4% paraformaldehyde (from 32% stock, 15714, Electron Microscopy Sciences) in phosphate-buffered saline (PBS) pH 7.4 (70011, Thermo Fisher). Three mice at postnatal days 100 ± 3 were used for hSOD1 aggregate staining. Three mice at postnatal days 66, 66 and 97 were used for p62 aggregate staining. The CNS was removed and fixed overnight in the same solution noted above. Whole brain and lumbar segments 4 and 5 were dissected, embedded in 4% (w/v) agar and sectioned on a vibratomb (Leica VT1000 S). 100 μm transverse spinal cord sections and sagittal whole brain sections were cut. Sections were blocked overnight in PBS with 10% donkey serum (D9553, Sigma) and 0.4% Triton X-100 (T8787, Sigma). Sections then were incubated at room temperature for two days in the above blocking buffer with primary antibodies [goat polyclonal anti-ChAT (1:100; AB144P, Millipore) and mouse monoclonal anti-SQSTM1/p62 (1:500; ab56416, Abcam) or mouse monoclonal anti-misfolded human SOD1 (1:250; MM-0072–02, MédiMabs]. After the primary incubation, six washes (>30 min each) in PBS with 0.4% Triton were followed by a one-day incubation at room temperature in the above wash buffer with donkey anti-mouse and anti-goat secondary antibodies (1.5 μg/mL; Jackson ImmunoResearch). After six more washes (as stated above), sections were mounted on microscope slides in Fluoromount G (OB100, ThermoFisher) using 100 μm spacers and allowed to set for >12 hr. Staining was visualized by confocal microscopy (Zeiss LSM 800).

### Quantification and statistical analysis

The quantitative data are expressed as mean ± SEM or mean ± SD of at least 3 (indicated in the figure legends, when the number was more than 3) independent experiments. The only exceptions are the polysome profiling experiment and the mass spectrometry experiment, which have two biological replicates. Statistical analysis was performed using ANOVA and student's $t$ test for data that are normally distributed or log-transformed after normality and variance homogeneity tests. Values of $p < 0.05$ were considered statistically significant.

## Acknowledgements

The authors would like to thank the Mazzoni lab members for constructive comments. We thank SL Pfaff (Salk Institute for Biological Studies, La Jolla, California, USA) for $Isl^{MN}$:GFP reporter mice; Brigitte Pettmann (Biogen, Cambridge, MA, USA) for instruction in SpMN dissection techniques; the Dana Farber Cancer Institute Flow Cytometry Facility, the Immunology Division Flow Cytometry Facility of Harvard Medical School, The Joslin Diabetes Center Flow Cytometry Core, Brigham and Women's Hospital Flow Cytometry Core, and Boston Children's Hospital Flow Cytometry Research

Facility for FACS isolation of primary motor neurons; AS Lee, AA Nugent, AP Tenney, MC Whitman, Engle laboratory members, and Project ALS consortium members for technical assistance and thoughtful discussion. We thank Seungkyu Lee for help in collecting single primary motor neurons. We also thank the Regev Lab/Klarman Cell Observatory at the Broad Institute for access to their next generation sequencing equipment. We thank Yi Fu for help in establishing hSOD1 transgenic ESC lines and thank Dr. L Giono for helping with the summary figure. This work was primarily supported by the PROJECT ALS grant A13-0416 to HW, ECE and EOM. Additional support from NYDH grant DOH01-C32243GG-3450000, MODBDF grant #5-FY14-99 and NICHD R01HD079682 to EOM. NIH/NINDS F31 NS 095571 and 103447 to JWS and DEI respectively. DA was partially supported by the NYU Dean's Dissertation award. RF was funded by Japan Heart Foundation/Bayer Yakuhin Research Grant Abroad and NIH Training grant in Genetics T32 GM007748. MFR was funded by NIH/NINDS (K08NS099502) and by NIH Training grants in HMS Genetics (NIGMS, T32-GM007748) and BWH Pathology (NHLBI, T32-HL007627). ECE is a Howard Hughes Medical Institute Investigator. CV acknowledges funding by the NIH/NIGMS (R01 GM113237 and 1R35GM127089-01), and the Zegar Family Foundation Fund for Genomics Research at New York University.

## Additional information

### Funding

| Funder | Grant reference number | Author |
|---|---|---|
| Project ALS | A13-0416 | Hynek Wichterle<br>Elizabeth C Engle<br>Esteban Orlando Mazzoni |
| Eunice Kennedy Shriver National Institute of Child Health and Human Development | R01HD079682 | Esteban Orlando Mazzoni |
| New York State Department of Health | DOH01-C32243GG-3450000 | Esteban Orlando Mazzoni |
| March of Dimes | March Of Dimes Birth Defects Foundation #5-FY14-99 | Esteban Orlando Mazzoni |
| National Institute of Neurological Disorders and Stroke | F31 NS 095571 | John W Smerdon |
| National Institute of Neurological Disorders and Stroke | F31 103447 | Dylan E Iannitelli |
| New York University | Dean's Dissertation award | Disi An |
| Japan Heart Foundation/Bayer Yakuhin Research Grant Abroad | | Ryosuke Fujiki |
| National Institutes of Health | T32 GM007748 | Ryosuke Fujiki |
| National Institute of Neurological Disorders and Stroke | | Matthew F Rose |
| National Institute of General Medical Sciences | T32-GM007748 | Matthew F Rose |
| National Heart, Lung, and Blood Institute | T32-HL007627) | Matthew F Rose |
| Howard Hughes Medical Institute | | Elizabeth C Engle |
| National Institute of General Medical Sciences | R01 GM113237 | Christine Vogel |
| National Institute of General Medical Sciences | 1R35GM127089-01) | Christine Vogel |
| Zegar Family Foundation Fund for Genomics Research at New York University | | Christine Vogel |

The funders had no role in study design, data collection and interpretation, or the decision to submit the work for publication.

## Author contributions

Disi An, Conceptualization, Resources, Data curation, Software, Formal analysis, Supervision, Validation, Investigation, Visualization, Methodology, Writing—original draft, Project administration, Writing—review and editing; Ryosuke Fujiki, Resources, Data curation, Formal analysis, Validation, Investigation, Methodology, Writing—review and editing; Dylan E Iannitelli, Conceptualization, Resources, Data curation, Software, Formal analysis, Validation, Investigation, Visualization, Methodology, Writing—review and editing; John W Smerdon, Resources, Data curation, Validation, Investigation, Methodology, Writing—review and editing; Shuvadeep Maity, Resources, Data curation, Software, Formal analysis, Validation, Investigation, Methodology, Writing—review and editing; Matthew F Rose, Conceptualization, Resources, Data curation, Software, Formal analysis, Supervision, Validation, Investigation, Methodology, Writing—review and editing; Alon Gelber, Investigation, Methodology; Elizabeth K Wanaselja, Joun Y Lee, Data curation, Writing—review and editing; Ilona Yagudayeva, Data curation, Investigation, Methodology, Writing—review and editing; Christine Vogel, Conceptualization, Formal analysis, Supervision, Methodology, Writing—review and editing; Hynek Wichterle, Elizabeth C Engle, Conceptualization, Supervision, Writing—review and editing; Esteban Orlando Mazzoni, Conceptualization, Supervision, Writing—original draft, Project administration, Writing—review and editing

## Author ORCIDs

Dylan E Iannitelli https://orcid.org/0000-0002-7654-9433
Shuvadeep Maity http://orcid.org/0000-0002-6031-4744
Matthew F Rose http://orcid.org/0000-0002-1148-4130
Christine Vogel http://orcid.org/0000-0002-2856-3118
Hynek Wichterle http://orcid.org/0000-0002-7817-0080
Esteban Orlando Mazzoni https://orcid.org/0000-0001-8994-681X

## Ethics

Animal experimentation: This study was performed in strict accordance with the recommendations in the Guide for the Care and Use of Laboratory Animals of the National Institutes of Health. Protocols were approved by Columbia University and Harvard University.

## Decision letter and Author response

Decision letter https://doi.org/10.7554/eLife.44423.025
Author response https://doi.org/10.7554/eLife.44423.026

# Additional files

## Supplementary files

• Supplementary file 1. List of plasmids.
DOI: https://doi.org/10.7554/eLife.44423.019
• Supplementary file 2. List of qPCR primers.
DOI:
• Transparent reporting form
DOI: https://doi.org/10.7554/eLife.44423.021

## Data availability

Sequencing data have been deposited in GEO under accession code GSE130938.

The following dataset was generated:

| Author(s) | Year | Dataset title | Dataset URL | Database and Identifier |
|---|---|---|---|---|
| An D, Fujiki R, Iannitelli DE, Smerdon JW, Maity S, Rose MF, Gelber A, Wanaselja EK, Yagudayeva I, Lee JY, Vogel C, Wichterle H, Engle EC, Mazzoni EO | 2019 | Sequencing data from Stem cell-derived cranial and spinal motor neurons reveal proteostatic differences between ALS resistant and sensitive motor neurons | http://www.ncbi.nlm.nih.gov/geo/query/acc.cgi?acc=GSE130938 | NCBI Gene Expression Omnibus, GSE130938 |

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
