## [Decision Letter]

Thank you for submitting your article "Stem cell-derived neurons reveal proteostatic differences between ALS resistant and sensitive motor neurons" for consideration by *eLife*. Your article has been reviewed by three peer reviewers, including Paola Arlotta as the Reviewing Editor and Reviewer #1, and the evaluation has been overseen by a Reviewing Editor and Marianne Bronner as the Senior Editor. The following individuals involved in review of your submission have agreed to reveal their identity: Justin Ichida (Reviewer #2); Brian Wringer (Reviewer #3).

The reviewers have discussed the reviews with one another and the Reviewing Editor has drafted this decision to help you prepare a revised submission.

Summary:

This is a strong, well-executed and highly impactful study reporting on a novel differential capacity by two classes of motor neurons to maintain a healthy proteasome and thus resist protein accumulation linked to ALS. Specifically, the authors report that CrMNs (which are more resistant to ALS) and spMNs (which are susceptible to disease) have different abilities to deal with protein accumulation and stress. Using both a novel in vitro ES-derived culture system for both CrMNs and SpMNs they demonstrate that ESC-derived CrMNs accumulate less human SOD1 and insoluble p62 than SpMNs. They go on to show that CrMNs have a stronger proteasome activity to degrade misfolded proteins and therefore are intrinsically more resistant to induced proteostatic stress than SpMNs. They confirm using optimized cultures of these primary neurons, isolated from the mouse early embryo, that endogenous cells also accumulate SOD1 differently and have different survival capacities connected to this.

All three reviewers agree that this is an outstanding study and make some suggestions to further strengthen the claims through revisions. I am summarizing hereafter the main points raised by the three reviewers, which the authors should mostly focus on as their manuscript is revised. In addition, I have appended the original comments to help understand the three specific suggestions.

1) There is general consensus that the study would benefit from a more extensive characterization of the identity of the ES-derived two classes of MNs. And, additionally, cells made in vitro should be compared to purified endogenous counterparts to more firmly establish their class identity. This is important because the core claims rest on the assumption that the two classes of MNs studied reflect the same classes in vivo. In comments by reviewers 1, 2 and 3 there are suggestions about the experiments that the authors may want to consider to resolve this issue, ideally using single cell RNA sequencing of ES-derived and in vivo purified MNs of the two classes.

2) There are key questions regarding the expression of SOD1 in the two cell types and how that relates to the phenotypes. It is important to determine if SOD1 is being produced at the same rate in iCrMNs and iSpMNs by examining translation rates. In addition, it is important to establish that insoluble SOD1 increases more dramatically in iSpMNs than iCrMNs after tunicamycin and/or CPA. Is there a way to verify that the proteostasis properties of iCrMNs and SpMNs are truly representative of their primary counterparts?

3) The authors should clarify how much the SOD1 overexpression and mutation are related to the phenotypes. Differences in terms of what effects result from SOD1 mutation (or overexpression) vs intrinsic vulnerability of lumbar motor neurons may be relevant to the disease.

*Reviewer #1:*

This is a very interesting, very well executed study reporting on a novel differential capacity by two classes of motor neurons to maintain a healthy proteasome and thus resisting protein accumulation linked to ALS. Specifically, they report that CrMNs (which are more resistant to ALS) and spMNs (which are susceptible to disease) have different abilities to deal with protein accumulation and stress. Using both a novel in vitro ES-derived culture system for both CrMNs and SpMNs they demonstrate that ESC-derived CrMNs accumulate less human SOD1 and insoluble p62 than SpMNs. They go on to show that CrMNs have a stronger proteasome activity to degrade misfolded proteins and therefore are intrinsically more resistant to induced proteostatic stress than SpMNs. They confirm using optimized cultures of these primary neurons, isolated from the mouse early embryo, that endogenous cells also accumulate SOD1 differently and have different survival capacities connected to this.

I find the work very interesting and only have some questions about the cellular models used, which I hope the reviewers can address.

1) In their in vitro cultures of primary cells, both neurons types (O/TrMNs and spMNs) were isolated very early, at E11.5. While this is needed to obtain neurons that can survive in culture (older neurons would not do so easily), it also brings up the issue that the two neuron types studied are at a different stage of maturation at the time of isolation. Given that SpMN are born earlier than O/TrMNs, is it possible that the differences observed are at least partly due to the fact that one cell type is younger than the other? Would cultures of E13 O/TrMNs (likely still quite viable in culture) give the same results?

2) How do the authors establish the class identity of the cells that they make from ESCs? Given that there are no single cell profiles of endogenous counterparts and comparisons are not made at the whole transcriptome level (for example by single cell RNAseq), there remain the issue that the cultures may contain cells that are not CrMNs, but rather cells that express a limited number of markers of this class. In addition, the cultures could be heterogenous and reflect different maturation states etc., possibly skewing the results/conclusions. A better characterization of cell identity would address this point.

With these considerations in mind, the paper is very interesting and on target for *eLife*.

*Reviewer #2:*

This impressive manuscript by Mazzoni and colleagues reveals an interesting possible explanation underlying the resistance of certain motor neuron subtypes to neurodegeneration in ALS. The authors provide evidence using primary and stem cell-derived spinal and cranial motor neurons to show that cranial motor neurons have a higher proteostatic capacity and more proteasome activity than spinal motor neurons, and that this leads to their ability to tolerate overexpression of mutant SOD1. Overall these findings provide an important conceptual advance for the field and most of the experiments are well-controlled. However, there are a few additional controls that would significantly strengthen the authors' conclusions.

Figure 1 – Although the authors previously published the NIP MN induction protocol (Mazzoni et al., 2013), it does not appear that the NIP-MNs were compared transcriptomically to primary CrMNs. Since much of the current manuscript's conclusions depend on the premise that iCrMNs are representative of CrMNs, it would be helpful to confirm this using RNA seq analysis of ESC-derived iCrMNs, iSpMNs, primary CrMNs, and primary SpMNs. The authors should already have the data for iCrMNs, iSpMNs, and primary SpMNs from their previous manuscript. If it is possible to collect enough primary CrMNs for RNA-seq, this would be a very helpful comparison.

Figure 3 – It is important to determine if SOD1 is being produced at the same rate in iCrMns and iSpMNs by examining translation rates using ribosome profiling or some other method.

Figure 3, Figure 4 and Figure 5 – One of the interesting points of the paper is that CrMNs and SpMNs may have different proteostasis capacities and different reliance on the proteasome vs. autophagy. The authors suggest that CrMNs have higher proteasomal activity while SpMNs presumably rely more on autophagy activity (or at least more autophagy than proteasome). To verify this, can the authors compare the levels of autophagy between SpMNs and CrMNs using an LC3-mRFP-GFP reporter or a similar assay?

Figure 3, Figure 4 and Figure 5 – The observed results could be explained by the possibility that MG-132 treatment increases autophagy in SpMNs but not CrMNs. The authors should determine if this is true or not.

Figure 6 and Figure 7 – There are reasons other than more severely impaired proteostasis that could cause tunicamycin or CPA to more profoundly affect iSpMN survival compared to iCrMN survival. The authors should confirm that insoluble SOD1 increases more dramatically in iSpMNs than iCrMNs after tunicamycin and/or CPA.

Figure 8 – It would greatly strengthen the conclusions if the authors could confirm using lentiviral overexpression or immunostaining of neurons from transgenic SOD1 mice that insoluble SOD1 increases more dramatically in primary SpMN cultures than primary CrMN cultures after tunicamycin, CPA, and MG-132. This would help to verify that the proteostasis properties of iCrMNs and SpMNs are truly representative of their primary counterparts.

*Reviewer #3:*

Mazzoni and colleagues perform a careful and thorough investigation into cell vulnerability in ALS. The parallel use of primary mouse and mouse ES-derived motor neurons adds substantial strength and validates the latter as a valuable tool. First, the authors isolate lumbar spinal motor versus oculomotor neurons and show that SOD1 aggregates are present in the former but not the latter. Isolation of these two populations using an Isl::GFP reporter reveals impaired in vitro survival of the spinal motor neurons. A similar result is observed using iNIL vs iNIP induction of cranial versus spinal motor neurons. Next, isogenic control lines harboring distinct SOD1 mutations are made, and these show similar capacity for differentiations. Compared to the iSpMNs, the ICrMNs have less SOD1 accumulation, particularly, the insoluble form. The authors then show that proteasome inhibition with MG-132 markedly increases the insoluble SOD1 in the SOD1 mutant iCrMNs. This finding is then extended to insoluble ubiquitinated proteins and p62, where the mutation in SOD1 seems less important, and MG-132 again gives a prominent and disproportionate increase in the iCMNs. Next, p62 increase is investigated more closely on cellular level in lumbar motor neurons, both for the induced SOD1 isogenics and primary mouse motor neurons. The iCrMNs are more resistant to Tunicamycin, CPA, and thapsigargin but not Brefeldin A, and a similar finding is shown for primary mouse motor neurons. Unbiased mass spec shows dysregulated 20S core subunits as well as higher proteasomal activity in the iCrMNs. Finally, proteasome activation using PD169316 boosts survival of the CPA-treated but not vehicle-treated iMNs, with the effect reaching significance in the iSpMNs.

This is an elegant and well-done study, with very nice confirmation between the induced motor neurons and the primary ones. While the authors show markers of resistant brainstem nuclei using the iNIP protocol, the percentages of cells marked by Pho2b and Sim2 are not shown. Some brainstem motor neurons, such as in the hypoglossal nucleus, are vulnerable and severely affected in ALS. Can further characterization of the iCrMN neurons clarify what percentage are oculomotor-like and whether some express markers of motor neurons that are not resistant? This is important because it clarifies a vulnerable vs non-vulnerable effect as opposed to simply a brainstem vs lumbar effect that covaries with vulnerability. Relatedly, is there information in the iSpMNs to compare in terms of lateral vs medial column identity?

A separate question is how much the SOD1 overexpression and mutation are related to the phenotypes. In the SOD1 results (Figure 3, Figure 4) the effects depend on the SOD1 mutation, whereas in Figure 5 for ubiquitin and p62 the effects largely do not appear to do so. It would be useful to know whether these results (Figure 5C and Figure 5E, Figure 6C, Figure 6F) also do not depend on SOD1 overexpression (ie seen in non-transgenic MNs as well, as mentioned in discussion). This is indirectly addressed in Figure 7 and Figure 9, which appear to use a non-transgenic line (although that does not appear to be clearly stated outside the discussion). Differences in terms of what effects result from SOD1 mutation (or overexpression) vs intrinsic vulnerability of lumbar motor neurons may be relevant to the fairly typical lumbar motor neuron-predominance of SOD1 ALS.

---

## [Author Response]

Summary:This is a strong, well-executed and highly impactful study reporting on a novel differential capacity by two classes of motor neurons to maintain a healthy proteasome and thus resist protein accumulation linked to ALS. Specifically, the authors report that CrMNs (which are more resistant to ALS) and spMNs (which are susceptible to disease) have different abilities to deal with protein accumulation and stress. Using both a novel in vitro ES-derived culture system for both CrMNs and SpMNs they demonstrate that ESC-derived CrMNs accumulate less human SOD1 and insoluble p62 than SpMNs. They go on to show that CrMNs have a stronger proteasome activity to degrade misfolded proteins and therefore are intrinsically more resistant to induced proteostatic stress than SpMNs. They confirm using optimized cultures of these primary neurons, isolated from the mouse early embryo, that endogenous cells also accumulate SOD1 differently and have different survival capacities connected to this.All three reviewers agree that this is an outstanding study and make some suggestions to further strengthen the claims through revisions. I am summarizing hereafter the main points raised by the three reviewers, which the authors should mostly focus on as their manuscript is revised. In addition, I have appended the original comments to help understand the three specific suggestions.

We would like to thank the reviewers for taking the time and effort to read our work and suggest improvements and are pleased that they find the work to be strong, well executed and highly impactful. While we provide a point-by-point rebuttal below, I would like to first summarize our response to the three primary points highlighted by the reviewers as well as additional new data we have included in the revised manuscript that we believe significantly strengthen the conclusion of the manuscript.

1) We provide single cell RNA-seq data from cultured primary rostral CrMN and SpMN and find a correlation between their gene expression. As expected, the in vitro derived-cells are not identical copies of those found in vivo. However, the cellular response to prostatic stress of ESC-derived motor neurons is predictive of the behavior of embryonically–derived ones. Thus, we believe induced motor neurons are an appropriate system to model that aspect of neuronal biology.

2) We have measured hSOD1 translation rates in iCrMNs and iSpMNs and found no significant difference.

3) We have investigated the effect of CPA stress on hSOD1 accumulation. While CPA induced more hSOD1 aggregation in both motor neuron types, iCrMNs still accumulated less insoluble hSOD1 than iSpMNs.

4) Our data support the conclusion that higher proteasome activity in CrMNs is partially responsible for their resistant phenotype. In the original version, we tested this hypothesis by activating the proteasome with a small molecule. However, small molecules could have off target effects. To test the hypothesis more specifically, we have now over-expressed PSMB5, one of the limiting proteasome subunits in postmitotic iSpMNs, and found improved iSpMN survival to CPA stress. We appreciate the additional time needed to include this additional experiment as we feel it significantly strengthens the manuscript.

1) There is general consensus that the study would benefit from a more extensive characterization of the identity of the ES-derived two classes of MNs. And, additionally, cells made in vitro should be compared to purified endogenous counterparts to more firmly establish their class identity. This is important because the core claims rest on the assumption that the two classes of MNs studied reflect the same classes in vivo. In comments by reviewers 1, 2 and 3 there are suggestions about the experiments that the authors may want to consider to resolve this issue, ideally using single cell RNA sequencing of ES-derived and in vivo purified MNs of the two classes.

As suggested by the reviewers, we addressed transcriptomic similarities between ESCderived MNs and primary MNs by single cell RNA sequencing of iCrMNs and iSpMNs cultured for 2, 5 and 7 days and E11.5 primary oculomotor and trochlear motor neurons (O/TrMNs) and SpMNs cultured for 7 days in similar medium.

Single cell RNA sequencing confirmed that CrMN and SpMN markers are differentially enriched in ESC-derived iCrMNs and iSpMNs (Figure 1—figure supplement 1D). ESC-derived and primary CrMNs and SpMNs all express motor neuron marker *Isl1*. CrMN fate markers *Phox2a* and *Phox2b* are enriched in iCrMNs and primary O/TrMNs, while SpMN fate markers *Lhx3* and *Mnx1/Hb9* are enriched in iSpMNs and primary SpMNs.

We compared all the samples together and clustered the cells using all significant principal components shown by T-distributed Stochastic Neighbor Embedding (t-SNE) plots in Figure 1H and Figure 1—figure supplement 1B. The clustering seems to be dominated by two factors: ESC versus primary origin and CrMN versus SpMN fate. About half of ESC-derived Day 5 and 7 iCrMNs clustered close to O/TrMNs on the second dimension, while the other half showed similar expression patterns with Day 5 and 7 iSpMNs. The heterogeneity might be explained by variance in differentiation and maturation speed, as well as in vitro culture stress that constantly selects for cells with higher resilience.

Principle component analysis (PCA) suggests that PC1 separates in vitro versus in vivo differentiated MNs (Figure 1—figure supplement 1C) and that PC2 separates O/TrMNs versus SpMNs (Figure 1I). Notably, the ESC-derived iCrMNs and iSpMNs cluster toward primary O/TrMNs and SpMNs, respectively, as they mature over time (Figure 1I). Day 7 iCrMNs clustered with primary O/TrMNs in the PC2 dimension.

In summary, the comparative transcriptome analysis further supports the notion that ESC-derived iCrMNs and iSpMNs resemble molecular features of the ALS-resistant O/TrMNs and ALS-sensitive SpMNs respectively.

We believe the field should be sincere on the identity of the cells we make and how to use them. The pressure to call ESC-derived cells “identical” to those in vivo is hurting the field by raising unnecessary expectations. As a field we should constantly thrive to improve differentiation outcomes. However, we should embrace differentiation strategies for their strength and be aware of and open about their limitations. Each model system has advantages and disadvantages. *Drosophila* is a wonderful model system, but perhaps a poor one to model endometrium-placenta vascularization. Thus, please allow us to be explicit about these ESC-derived cells.

What these induced motor neurons are:

Induced CrMN and SpMN (iCrMN and iSpMN) look like neurons and express genes associated with neuronal and motor neuron identity. iCrMNs express *Phox2a, Phox2b* and markers associated with rostral CrMN fate. They lack *Lhx3* and *Hb9*, which are expressed by SpMNs and some more caudal CrMNs. Thus, both iCrMNs and iSpMNs express cardinal markers of cranial and spinal motor neuron fate.

The iCrMN versus iSpMN approach we take in this manuscript successfully models at least one cellular difference between ALS-resistant and ALS-sensitive motor neurons and suggests higher proteasome activity as a possible mechanism of resistance to ALS induced neurodegeneration. We should note that this might not be the only difference. A recent preprint utilizing a slightly different strategy but also forcing Phox2a expression reports that stem cell-derived cranial motor neurons are more resistant to kainic acid toxicity. We have included this new development in the Discussion section of the revised manuscript.

What these neuros are not:

Based on global gene expression profile and the differentiation protocol, neither of these ESC-derived motor neurons are of a unique motor neuron pool identity. This is why we named iCrMNs and iSpMNs as “rostral cranial motor neurons” and “spinal motor neurons”, instead of oculomotor neurons and cervical spinal motor neurons, respectively. We believe iCrMN identity is consistent with rostral CrMN fate but are reluctant to call them oculomotor neurons.

2) There are key questions regarding the expression of SOD1 in the two cell types and how that relates to the phenotypes. It is important to determine if SOD1 is being produced at the same rate in iCrMns and iSpMNs by examining translation rates.

To measure hSOD1 translation efficiency, we performed polysome profiling in hSOD1 A4V iCrMNs and iSpMNs. Small (40S) and large (60S) ribosome subunits, monosomes (80S) and polysomes are shown as different peaks in the polysome profiles and separated into different fractions accordingly (Figure 3—figure supplement 1A and B). RNA was extracted from the whole cell lysate and polysome fractions. The ratios between polysome-associated hSOD1 A4V mRNA and whole hSOD1 A4V mRNA were similar between iCrMNs and iSpMNs, which was quantified by RT-qPCR and normalized by ribosomal protein RPL19 (Figure 3—figure supplement 1C). These data suggest that the hSOD1 transgene is translated at similar rates in iCrMNs and iSpMNs and that the differential accumulation of hSOD1 mutant proteins between them is probably because of differential degradation.

Additionally, we should note that a more efficient hSOD1 translation is unlikely to explain why iSpMNs contain more insoluble hSOD1 aggregates during maturation. When reducing proteasome activity, iSpMNs revert the trend and accumulate less than iCrMNs.

In addition, it is important to establish that insoluble SOD1 increases more dramatically in iSpMNs than iCrMNs after tunicamycin and/or CPA.

To test the sensitivity of protein aggregation to additional proteostatic stress in iCrMNs and iSpMNs, we treated the MNs expressing hSOD1 A4V with DMSO or 20 µM CPA from Day 2 to Day 5, similar to the survival assays. The accumulation of hSOD1 A4V was quantified on Day 5 between CPA and DMSO treated cells by western blot (Figure 7—figure supplement 2a).

CPA treated iCrMNs and iSpMNs accumulated ~3.5-fold more insoluble hSOD1 A4V compared to DMSO treated ones (Figure 7—figure supplement 2B). There was no significant difference between iCrMNs and iSpMNs in terms of increasing hSOD1 accumulation in response to CPA.

Notably, iCrMNs still accumulated less insoluble hSOD1 A4V than iSpMNs after 3-day 20 µM CPA treatment on Day 5 (Figure 7—figure supplement 2C). Taken together, these data suggest that additional proteostatic stress accelerate the accumulation of insoluble hSOD1 mutant proteins and that iCrMNs maintain the superior capability to deal with misfolding proteins under CPA treatment.

Is there a way to verify that the proteostasis properties of iCrMNs and SpMNs are truly representative of their primary counterparts?

To address this question, we have directly tested if the differential proteostatic properties of iCrMNs and iSpMNs are conserved in primary cultures (Figure 8). iCrMNs, iSpMNs, primary rostral CrMNs (oculomotor and trochlear nuclei), and primary SpMNs were treated with the same CPA and tunicamycin doses. Similar to iCrMNs versus iSpMNs, rostral primary CrMNs survived better than primary SpMNs. Thus, the ESC-derived MNs are representative of their in vivo counterparts.

Moreover, differential accumulation of hSOD1- and p62-containing aggregates between O/TrMNs and SpMNs in *SOD1*^G93A^ mutant mice supports the hypothesis that primary rostral CrMNs deal better with ALS-related misfolding proteins (Figure 1A and B, Figure 6D and E).

Taken together, we first used ALS-associated hSOD1 mutant proteins as an example of ALS-relevant proteostatic stress to reveal the differential proteostatic properties between CrMNs and SpMNs in vitro and in vivo. We then established the inherent differential proteostatic capacity using the non-transgenic ESC-derived and primary MNs, which probably explains why CrMNs are resistant to both familial and sporadic ALS.

3) The authors should clarify how much the SOD1 overexpression and mutation are related to the phenotypes. Differences in terms of what effects result from SOD1 mutation (or overexpression) vs intrinsic vulnerability of lumbar motor neurons may be relevant to the disease.

The specific effects of SOD1 mutations A4V and G93A are addressed by two results in Figure 3 and Figure 4. As pointed out by reviewer #3, there is differential accumulation of insoluble hSOD1 A4V and G93A but not hSOD1 WT after long-term culture (lower in iCrMNs) and after MG-132 treatment (higher in iCrMNs). The *SOD1*^G93A^ mutant mice also accumulate less hSOD1 and p62 aggregates in O/TrMNs than in lumbar SpMNs (Figure 1A and B, Figure 6D and E). Thus, hSOD1 carrying ALS mutations behave differently than WT hSOD1.

The survival assays with CPA and tunicamycin treatment were performed in non-transgenic ESC-derived and primary CrMNs and SpMNs and we have now clarified this in the revised Results section. These data clearly demonstrate that CrMNs are inherently more resistant to proteostatic stress than SpMNs (Figure 7 and Figure 8). Finally, the small molecule (PD169316) (Figure 10D) and genetic (PSMB5) (Figure 10F) rescue of iSpMN sensitivity to CPA were also performed in non-transgenic lines. Thus, sensitivity to proteostatic stress and its rescue are intrinsic motor neuron properties.

Reviewer #1:[…] I find the work very interesting and only have some questions about the cellular models used, which I hope the reviewers can address.1) In their in vitro cultures of primary cells, both neurons types (O/TrMNs and spMNs) were isolated very early, at E11.5. While this is needed to obtain neurons that can survive in culture (older neurons would not do so easily), it also brings up the issue that the two neuron types studied are at a different stage of maturation at the time of isolation. Given that SpMN are born earlier than O/TrMNs, is it possible that the differences observed are at least partly due to the fact that one cell type is younger than the other? Would cultures of E13 O/TrMNs (likely still quite viable in culture) give the same results?

We are also aware of this difference of maturation between isolated primary O/TrMNs and SpMNs and agree that it deserves future investigation. Unfortunately, because of technical difficulties of dissection at later stages (changes in MN marker Isl1-GFP transgene expression and dissection difficulties), we are not able to answer this question with the current mice. We believe, however, that our in vivo experiments address some of these concerns. Differential accumulation of hSOD1 and p62 aggregates between O/TrMNs and SpMNs were identified in *SOD1*^G93A^ mutant mice (Figure 1A and B, Figure 6D and E). Thus, small initial maturation differences should be a minimal contributing factor more than two months after the neurons were born.

2) How do the authors establish the class identity of the cells that they make from ESCs? Given that there are no single cell profiles of endogenous counterparts and comparisons are not made at the whole transcriptome level (for example by single cell RNAseq), there remain the issue that the cultures may contain cells that are not CrMNs, but rather cells that express a limited number of markers of this class. In addition, the cultures could be heterogenous and reflect different maturation states etc., possibly skewing the results/conclusions. A better characterization of cell identity would address this point.

We agree that it is important to further characterize the identity of ESC-derived MNs. We have addressed this question by single cell RNA sequencing. Please see the response to the general comment 1 above.

With these considerations in mind, the paper is very interesting and on target for eLife.Reviewer #2:[…] However, there are a few additional controls that would significantly strengthen the authors' conclusions.Figure 1 – Although the authors previously published the NIP MN induction protocol (Mazzoni et al., 2013), it does not appear that the NIP-MNs were compared transcriptomically to primary CrMNs. Since much of the current manuscript's conclusions depend on the premise that iCrMNs are representative of CrMNs, it would be helpful to confirm this using RNA seq analysis of ESC-derived iCrMNs, iSpMNs, primary CrMNs, and primary SpMNs. The authors should already have the data for iCrMNs, iSpMNs, and primary SpMNs from their previous manuscript. If it is possible to collect enough primary CrMNs for RNA-seq, this would be a very helpful comparison.

We agree that it is important to further characterize the identity of ESC-derived MNs. We have addressed this question by single cell RNA sequencing. Please see the response to the general comment 1 above.

Figure 3 – It is important to determine if SOD1 is being produced at the same rate in iCrMns and iSpMNs by examining translation rates using ribosome profiling or some other method.

We have addressed this question by polysome profiling and quantification of polysome associated hSOD1 mRNA versus whole hSOD1 mRNA by RT-qPCR. Please see the response to the first question of general comment 2 above.

Figure 3, Figure 4 and Figure 5 – One of the interesting points of the paper is that CrMNs and SpMNs may have different proteostasis capacities and different reliance on the proteasome vs. autophagy. The authors suggest that CrMNs have higher proteasomal activity while SpMNs presumably rely more on autophagy activity (or at least more autophagy than proteasome). To verify this, can the authors compare the levels of autophagy between SpMNs and CrMNs using an LC3-mRFP-GFP reporter or a similar assay?

We agree with the reviewer that SpMNs should rely on a different protein degradation machinery, very likely the autophagy pathway.

We inhibited autophagy by 3-Methyladenine and Bafilomycin A1 and measured the level of insoluble hSOD1 by western blot. No significant effects were detected following treatment with autophagy inhibitors. It is always difficult to extract significant conclusions from negative results. Thus, we cannot conclude autophagy is not a contributing factor to the control of insoluble protein aggregation. We have changed the relevant paragraph in the Discussion section.

Figure 3, Figure 4 and Figure 5 – The observed results could be explained by the possibility that MG-132 treatment increases autophagy in SpMNs but not CrMNs. The authors should determine if this is true or not.

To test if autophagy affects the differential aggregate accumulation during MG-132 treatment, we co-treated the MNs with MG-132 and autophagy inhibitors 3-

Methyladenine, Bafilomycin A1 or both. No significant changes were observed on the differential accumulation of insoluble hSOD1 compared to MG-132 treatment alone. Thus, autophagy does not seem to be responsible for the phenotypes observed with MG-132 treatment.

Figure 6 and Figure 7 – There are reasons other than more severely impaired proteostasis that could cause tunicamycin or CPA to more profoundly affect iSpMN survival compared to iCrMN survival. The authors should confirm that insoluble SOD1 increases more dramatically in iSpMNs than iCrMNs after tunicamycin and/or CPA.

Please see the response to the second question of general comment 2 above.

Figure 8 – It would greatly strengthen the conclusions if the authors could confirm using lentiviral overexpression or immunostaining of neurons from transgenic SOD1 mice that insoluble SOD1 increases more dramatically in primary SpMN cultures than primary CrMN cultures after tunicamycin, CPA, and MG-132. This would help to verify that the proteostasis properties of iCrMNs and SpMNs are truly representative of their primary counterparts.

We agree with the reviewer that this experiment would strengthen the conclusion. Unfortunately, we are not able to conduct the experiment because of technical difficulties. The presented CPA and tunicamycin experiments required a large number of animals and dissection time from WT animals. We have tried in vain to obtain and maintain a good number of healthy primary oculomotor neurons from the *SOD1*^G93A^ mouse embryos.

Reviewer #3:[…] This is an elegant and well-done study, with very nice confirmation between the induced motor neurons and the primary ones. While the authors show markers of resistant brainstem nuclei using the iNIP protocol, the percentages of cells marked by Pho2b and Sim2 are not shown. Some brainstem motor neurons, such as in the hypoglossal nucleus, are vulnerable and severely affected in ALS. Can further characterization of the iCrMN neurons clarify what percentage are occulomotor-like and whether some express markers of motor neurons that are not resistant? This is important because it clarifies a vulnerable vs non-vulnerable effect as opposed to simply a brainstem vs lumbar effect that covaries with vulnerability. Relatedly, is there information in the iSpMNs to compare in terms of lateral vs medial column identity?

We agree that it is important to further characterize the identity of ESC-derived MNs. We have addressed this question by single cell RNA sequencing. Please see the response to the general comment 1 above.

Since *Lhx3* is used to program iSpMN fate, the iSpMN should mimic the identity of medial motor neuron column. However, as explained in the answer to the general comment 1 above, iSpMNs do not have a specific rostro-caudal identity.

A separate question is how much the SOD1 overexpression and mutation are related to the phenotypes. In the SOD1 results (Figures3, Figure 4) the effects depend on the SOD1 mutation, whereas in Figure 5 for ubiquitin and p62 the effects largely do not appear to do so. It would be useful to know whether these results (Figure 5C and Figure 5E, Figure 6C, Figure 6F) also do not depend on SOD1 overexpression (ie seen in non-transgenic MNs as well, as mentioned in discussion). This is indirectly addressed in Figure 7 and Figure 9, which appear to use a non-transgenic line (although that does not appear to be clearly stated outside the discussion). Differences in terms of what effects result from SOD1 mutation (or overexpression) vs intrinsic vulnerability of lumbar motor neurons may be relevant to the fairly typical lumbar motor neuron-predominance of SOD1 ALS.

Please see the response to the general comment 3 above.